# *Staphylococcus aureus* functional amyloids catalyze degradation of β-lactam antibiotics

Elad Arad [1,2], Kasper B. Pedersen [3], Orit Malka[2], Sisira Mambram Kunnath[1,2], Nimrod Golan[4], Polina Aibinder[5], Birgit Schiøtt [3,6], Hanna Rapaport[1,5], Meytal Landau[4,7] & Raz Jelinek [1,2] ✉

Antibiotic resistance of bacteria is considered one of the most alarming developments in modern medicine. While varied pathways for bacteria acquiring antibiotic resistance have been identified, there still are open questions concerning the mechanisms underlying resistance. Here, we show that alpha phenol-soluble modulins (PSMαs), functional bacterial amyloids secreted by *Staphylococcus aureus*, catalyze hydrolysis of β-lactams, a prominent class of antibiotic compounds. Specifically, we show that PSMα2 and, particularly, PSMα3 catalyze hydrolysis of the amide-like bond of the four membered β-lactam ring of nitrocefin, an antibiotic β-lactam surrogate. Examination of the catalytic activities of several PSMα3 variants allowed mapping of the active sites on the amyloid fibrils' surface, specifically underscoring the key roles of the cross-α fibril organization, and the combined electrostatic and nucleophilic functions of the lysine arrays. Molecular dynamics simulations further illuminate the structural features of β-lactam association upon the fibril surface. Complementary experimental data underscore the generality of the functional amyloid-mediated catalytic phenomenon, demonstrating hydrolysis of clinically employed β-lactams by PSMα3 fibrils, and illustrating antibiotic degradation in actual *S. aureus* biofilms and live bacteria environments. Overall, this study unveils functional amyloids as catalytic agents inducing degradation of β-lactam antibiotics, underlying possible antibiotic resistance mechanisms associated with bacterial biofilms.

Antibiotic resistance by pathogenic bacteria is among the most pressing challenges in modern medicine and healthcare[1]. Indeed, pathogenic bacteria and other microorganisms have been found to employ various mechanisms to fight antibiotic potency, including genetic mutations, evolving specific receptors targeting antibiotics which activate cascades of enzymatic antibiotic-degradation[2], and formation of physical protective barriers which adsorb or block antibiotic molecules from reaching their targets[3]. Methicillin-resistant *Staphylococcus Aureus* (MRSA), in particular, is considered one of the most health threatening antibiotic-resistant pathogens[2,4,5]. Antibiotic

[1]Ilse Katz Institute (IKI) for Nanoscale Science and Technology, Ben Gurion University of the Negev, Beer Sheva 8410501, Israel. [2]Department of Chemistry, Ben Gurion University of the Negev, Beer Sheva 8410501, Israel. [3]Department of Chemistry, Aarhus University, Langelandsgade 140, 8000 Aarhus C, Denmark. [4]Department of Biology, Technion-Israel Institute of Technology, Haifa 3200003, Israel. [5]Avram and Stella Goldstein-Goren Department of Biotechnology Engineering, Ben Gurion University of the Negev, Beer Sheva 8410501, Israel. [6]Interdisciplinary Nanoscience Center (iNANO), Aarhus University, Gustav Wieds Vej 14, 8000 Aarhus C, Denmark. [7]Centre for Structural Systems Biology (CSSB), and European Molecular Biology Laboratory (EMBL), Hamburg 22607, Germany. ✉e-mail: Razj@bgu.ac.il

resistance by *S. aureus* have been traced to alteration of target proteins' binding sites[6]. Another potent resistance strategy is the generation of antibiotic-degrading enzymes such as β-lactamases which degrade β-lactam antibiotics[7]. Indeed, antibiotic resistance is particularly acute in the case of β-lactams, which include penicillin, penems, and others[8], that are among the most commonly-used drugs worldwide.

Bacterial biofilms, the rigid matrix produced by many pathogenic bacteria, also play a role in antibiotic resistance as biofilm layers have been shown to adsorb small molecules thereby putatively blocking access to the bacterial colonies[9]. Biofilm frameworks comprise varied biomolecules, major among them are protein amyloid fibrils[10]. Amyloid fibrils, observed in diverse protein systems and mostly associated with neurodegenerative diseases[11], have been also increasingly encountered in bacterial biology. Functional bacterial amyloids (referred to as FuBAs) serve as building blocks for biofilm scaffolding[12], facilitate resistance to stress conditions[13], act as bio-adhesives anchoring agents[14], create channels in the biofilm matrix for nutrient uptake and transport[15,16], and even act as antibacterial agents, achieving dominance over other microbial species[17–19]. While prominent FuBAs, including *curli* (secreted by *E. coli*)[20,21], *Fap (P. aeruginosa)*[8], and *Tas (bacillus)*[16] adopt β-sheet amyloid structures, amyloid proteins produced by *S. aureus*, specifically belonging to the *phenol-soluble modulin* (PSM) family have been shown to form unique cross-α organizations[22,23]. Specifically, the PSMα peptides which are highly amphiphilic, induce the formation of α-helical bilayers that bury their hydrophobic residues and expose the hydrophilic ones[24].

Amyloidogenic proteins have been shown to exhibit catalytic properties[25–28]. Self-assembled peptides were reported to catalyze hydrolysis, condensation, redox reactions[26,27], and even multi-step cascade-reactions[29,30]. While almost all previous studies have focused on synthetic *de-novo* peptides or amyloidogenic protein domains as catalytic agents[31–35], we have recently demonstrated that native, physiological amyloids can catalyze disease-related reactions and key metabolic reactions[36,37]. These discoveries suggest that amyloid proteins may act as catalysts for varied biological reactions and functional targets. Here, we demonstrate that members of the PSMα family, particularly PSMα3, catalyze hydrolysis of the amide like bond of the four membered β-lactam ring antibiotics. Through structure/function analyses of PSMα3 variants, putative catalytic sites on the amyloid fibrils' surface are localized, linked to both the cross-α organization of the amyloid fibrils and electrostatic interactions between the β-lactam molecules and lysine sidechains. Amyloid-catalyzed β-lactam degradation may underscore an intriguing paradigm accounting for antibiotic resistance of *S. aureus*, and pathogenic bacteria in general.

## Results and discussion

The hypothesis and experimental scheme underlining this study are illustrated in Fig. 1a. PSMα amyloid fibrils (physiologically identified as constituents of *S. aueureus* biofilms, and shown to be secreted by this bacterial species as toxins[19,38]) adsorb β-lactams which are subsequently degraded through catalyzing hydrolysis of the 4-member β-lactam ring. Structurally, β-lactams contain an electrophilic, activated-amide bond ring, allowing it to react with various enzyme nucleophiles leading to bacterial cell destruction[8]. The experiments presented below are designed to examine whether PSMα amyloids, through active sites upon the amyloid fibril surface (schematically represented as "scissors"), catalyze cleavage of the four-membered β-lactam ring, rendering the molecule functionally inactive. Specifically, we tested PSMα amyloid-mediated degradation of nitrocefin, a widely-studied β-lactam surrogate of penicillin[39].

Figure 1b−e presents structural analyses of the PSMα1−4 amyloid fibrils investigated. Figure 1b depicts the amino acid sequences of the peptides, highlighting in different colors the amyloid surface-displayed residues which may play roles in catalysis, and the formal

charges of the sequences. Specifically, acidic residues (red), basic residues (blue), or neutral nucleophilic residues (turquoise) have been observed in enzymatic active sites[40–42]. Particularly, in many enzymes, polar residues in the active sites play prominent functions by binding substrate molecules, activating proximate residues, or partaking in nucleophilic attacks[42] (similar catalytic properties were reported in the case of β-sheet forming peptides[43]).

Fluorescence emission measurements of PSMα fibrillar samples incubated with the amyloid-specific dye *Amytracker*−680, which fluorescence intensity correlates with repeating amyloidogenic and positively-charged amyloid domains[44,45], are portrayed in Fig. 1d. The bar diagram in Fig. 1d indicates that variability in amyloid formation by the PSMα assemblies was apparent. PSMα2 and PSMα3 gave rise to high fluorescence of the dye, indicating pronounced amyloid organization, while PSMα1 and PSMα4 generated lesser *Amytracker*−680 fluorescence, reflecting lesser amyloid fibril organization of the peptide aggregates. The circular dichroism (CD) analysis in Fig. 1e further illuminates the secondary structures of the PSMα peptides. The CD spectra of PSMα1, PSMα2, and PSMα3 indicate that the three peptides adopted α-helical structures in solution, reflected in the maxima at 195 nm and minima at 210 nm and 225 nm[23,46], and consistent with the total reflectance-FTIR (ATR-FTIR) amide-I spectroscopy showing a peak in 1652 cm$^{-1}$ (Fig. S1). PSMα4, on the other hand, featured β-sheet assemblies, accounting for the spectral minimum in 215 nm and peak at 198 nm, consistent with formation of cross-β fibril organization[22,47], and supported by the observation of the ATR-FTIR amide-I peak in 1625 cm$^{-1}$ (Fig. S1).

Figure 2 explores PSMα-mediated catalysis of β-lactam ring breakup via amide bond hydrolysis. In the experiments, we monitored the four-membered ring opening in nitrocefin following addition of the PSMα amyloids (pre-incubated at concentration of 400 μM in water for two hours and then buffered with Hepes, pH 7.4) and tracking the absorbance at 480 nm[39]. Figure 2a reveals significantly different catalytic properties of the PSMα aggregates. Specifically, PSMα2 and PSMα3 amyloids markedly accelerated nitrocefin degradation, resulting in almost complete turnover (almost 100% yield) in the case of PSMα2 and 75% for PSMα3. PSMα1 gave rise to lower yield of ~50%, while PSMα4 induced minor degradation of nitrocefin (yield of ~15%). A control experiment attested to very low β-lactam degradation by β-amyloid fibrils, previously reported to catalyze ester-hydrolysis and catechol oxidation reactions[36], even at high fibrils concentration (Fig. S2). Similarly, the cross-β *Pseudomonas Aeruginosa*-originated bacterial-amyloids FapB and FapC[48] induced minor nitrocefin degradation, displaying low reaction rates and yield (Fig. S3).

Figure 2b depicts the initial nitrocefin degradation reaction rates, $V_0$ (calculated from linear fitting of the degraded-nitrocefin absorbance slopes, Fig. 2a, in the initial 15 min), as a function of PSMα peptide concentrations (initial nitrocefin concentration 60 μM). Consistent with the kinetic data in Fig. 2a, PSMα2 amyloids exhibited the highest $V_0$ followed by PSMα3 aggregates, while PSMα1 and PSMα4 exhibited lesser catalytic activities (i.e., low initial degradation of nitrocefin). Figure 2b also shows that the $V_0$ for all peptides exhibited linear dependency reflecting higher catalytic activity when greater amyloid concentrations were present in the aqueous solution. Importantly, nitrocefin breakup was observed only above a ~30 μM threshold (Fig. 2b, inset). Indeed, we observed that below this threshold concentration all four PSMα amyloids were practically soluble (Fig. S4). Importantly, PSMα2 and PSMα3 amyloid fibrils remain active and recyclable after saturation with the nitrocefin substrate (Fig. S5). Overall, Fig. 2b attests to a direct relationship between PSMα amyloid formation and nitrocefin degradation.

Measurements of $V_0$ as function of initial substrate (nitrocefin) concentrations (keeping the PSMα concentration constant at 170 μM, assuring amyloid formation, e.g., above the 30 μM threshold) are

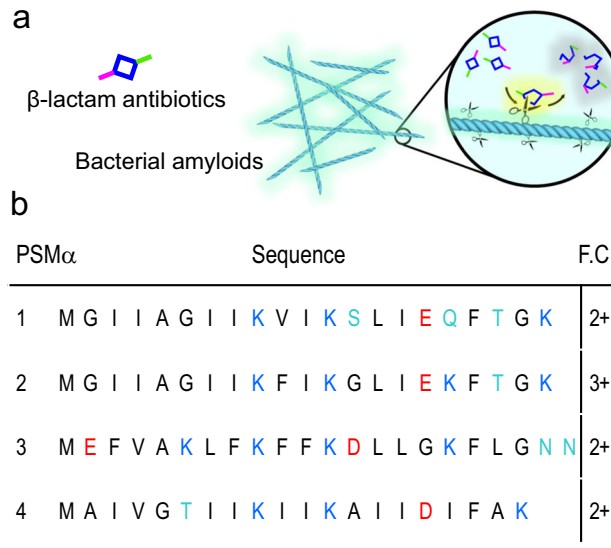

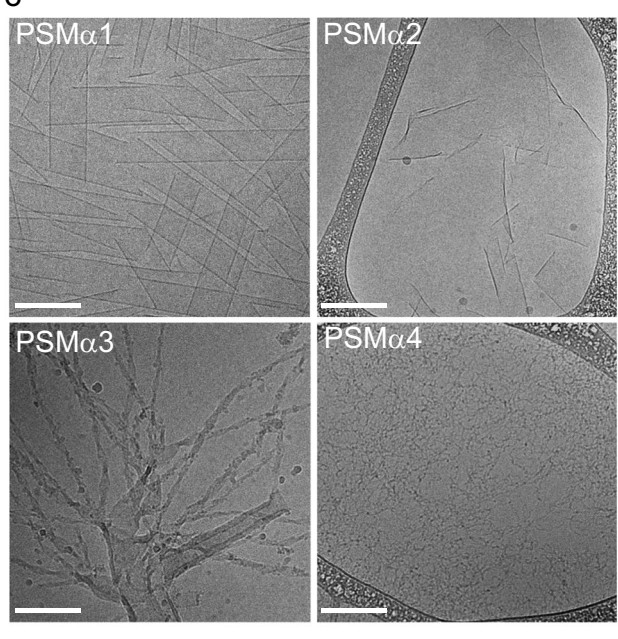

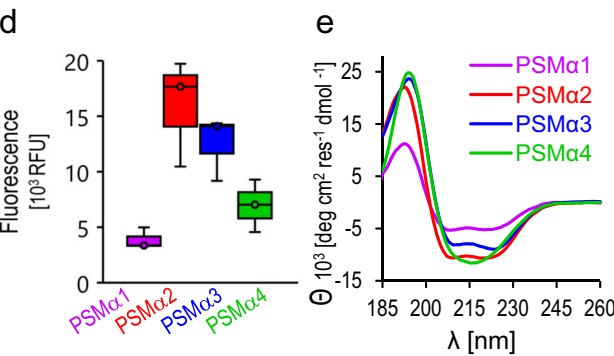

**Fig. 1 | S. aureus-secreted functional amyloids: Hypothesis and structural characterization of the PSMα amyloids. a** PSMα amyloid fibrils bind β-lactams and catalyze their degradation through amide bond hydrolysis. **b** Sequences of PSMα 1–4. Anionic, cationic, and nucleophilic residues are marked in red, blue, and turquoise, respectively. The formal charge (F.C.) of each sequence is noted on the right. **c** Cryo-TEM images of the peptides (concentration 300 μM). Bars correspond to 200 nm. All PSMα samples were pre-incubated in water for two hours and buffered with Hepes prior to image acquisition. **d** Amyloid staining with *Amytracker 680* (excitation 552 nm, emission 654 nm, PSMα concentrations were 300 μM). Data presented in box-and-whisker plot, with mean line and quartile calculation using inclusive median, $N = 3$. **e** Circular dichroism (CD) spectra of the peptides (concentration 170 μM, peptides dissolved in Hepes buffer, pH = 7.4). Source data are provided as a Source Data file.

the MM-constant ($K_{cat}/K_M$), considered a measure for both substrate binding affinity and conversion kinetics into product. Consistent with the data in Fig. 2a–c, the bar diagram in Fig. 2d demonstrates that PSMα2 and PSMα3 exhibit significantly higher catalytic efficiencies compared to PSMα1 and PSMα4.

To decipher the structural parameters and putative active sites in the most catalytically efficient peptide amyloid - PSMα3, we analyzed the structural and catalytic properties of several PSMα3 variants, either displaying altered structural and functional features (Fig. 3) or exhibiting strategic point mutations (Fig. 4). Figure 3a shows sequences of the PSMα3 variants tested. *α3-cationic* has pronounced positive charge since all acidic functional groups were amidated (the glutamic acid in position 2, aspartic acid in position 13 and asparagine in the C-terminus), thereby avoiding possible electrostatic repulsion with the anionic nitrocefin substrate. In comparison, the lysine residues in the *α3-EG/K* variant were capped with two units of ethylene-glycol, thus blocking the positive residue and increasing the amphiphilicity of the side-chains. *α3-F10P* does not adopt cross-α architecture due to the rigid turn-motif introduced by substitution of phenylalanine with proline[22]. Similar disruption of cross-α organization was designed in the case of the *α3-(d)FF* variant, in which the overall sequence is retained but the L-Phe residues in positions 10 and 11 were substituted with diastereomeric D-Phe[22]. Another control PSMα3 variant was a scrambled sequence *(α3-scrambled)*. This variant contains the same amino acids as the parent peptide, however it does not display the helical amphipathic pattern important for α-helical folding[50].

Figure 3b shows the cryo-TEM morphologies of the PSMα3 variants following incubation for two hours in water and buffering with Hepes pH 7.4. Different than native PSMα3 assembled primarily as twisted fibrils, the *α3-cationic* variant produced micron-scale thin sheets, ascribed to pronounced effect of the electrostatic repulsion between the positive moieties, blocking sheet-to-fibril transformations[51]. Surprisingly, both *α3-scrambled* and *α3-(d)FF* appear to form thin fibrillar structures, accounting for the amphiphilic sequences of these variants and adoption of β-sheet structure. In contrast to the other variants, the cryo-TEM images in Fig. 3b reveal that *α3-F10P* and *α3-EG/K* formed amorphous aggregates. In the case of *α3-F10P*, this result is likely due to the structural rigidity introduced by proline-10, disrupting folding neither into α-helices or β-sheets. The amorphous structures of *α3-EG/K* likely reflect the effects of the ethylene-glycol units decorating the four lysine residues, blocking participation of the charged lysines in peptide self-assembly. Fluorescent labeling of these variants with *Amytracker-680* is consistent with the TEM images (Fig. S6).

The attenuated total reflectance-FTIR (ATR-FTIR) spectra of the amide-I vibration region of the PSMα3 variants in Fig. 3c account for the secondary structures of the peptides[52]. Figure 3c demonstrates that native PSMα3 and the *α3-cationic* variant gave rise to similar peaks centered at 1655 cm$^{-1}$, the fingerprint peak for α-helical structure[53]. *α3-F10P* also gave rise to a signal at 1647–1650 cm$^{-1}$, although markedly broader, accounting for substantial degree of unordered random-coil

depicted in Fig. 2c. Notably, the shapes of the four curves (accounting for the respective PSMα peptides) indicate a Michaelis-Menten (MM) kinetic dependence, corroborating an enzyme-like catalytic activity[49]. Table S1 summarizes the pertinent enzymatic parameters extracted from the initial reaction rate data in Fig. 2c using non-linear regression fit to a MM reaction model[49]. Figure 2d presents the catalytic efficiencies, calculated for each peptide by dividing the turnover rate by

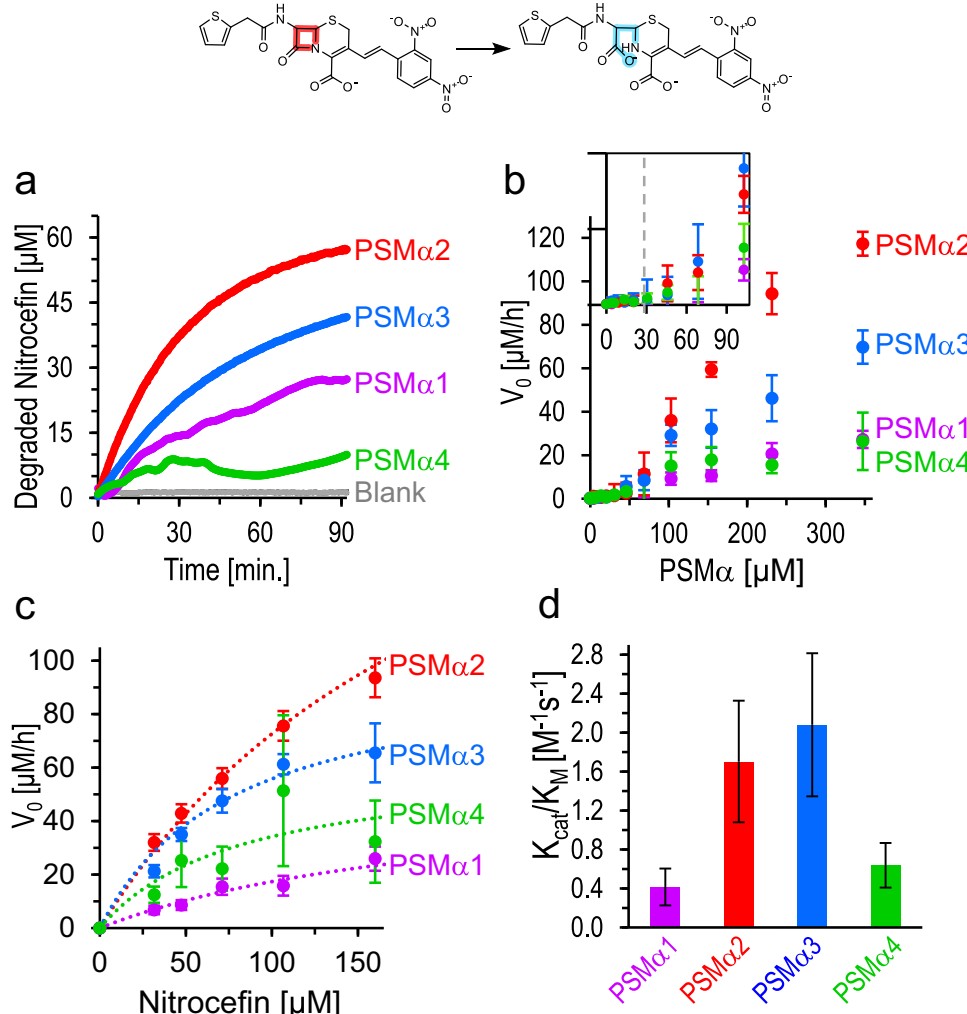

**Fig. 2 | Catalytic amide bond hydrolysis of nitrocefin by the PSMα assemblies.** A scheme of nitrocefin β-lactam ring opening and degradation (top row). **a** Graph depicting degradation of nitrocefin (initial nitrocefin concentration 60 μM) in the presence of PSMα assemblies (350 μM). **b** Initial nitrocefin-degradation reaction rate, $V_0$, as a function of initial preassembled-PSMα concentration (the peptides were pre-incubated in water for two hours and then buffered with Hepes prior to mixing with nitrocefin at concentration of 60 μM). The inset depicts the low-concentration peptide region, underscoring a negligible catalytic activity at $V_0$ lower than 30 μM. The values are represented as average ± SEM, $N = 3$. **c** Initial nitrocefin-degradation reaction rate as function of initial nitrocefin concentration (PSMα peptide concentrations of 170 μM). Each point represents an average ± SEM, $N_{PSMα1} = 8$, $N_{PSMα2} = 19$, $N_{PSMα3} = 10$, $N_{PSMα4} = 5$. The dotted line is the fitting to a Michaelis-Menten (MM) model using non-linear regression. **d** The catalytic efficiency (ε) is derived from the fitting to MM model (using the catalytic parameters in Table S1). The values represented as the calculated ratio ($K_{cat}/K_M$) and the error bars are the confidence interval, derived from the fitting to the model. Source data are provided as a Source Data file.

organization[54]. The FTIR spectrum of *α3-(d)FF* was significantly different than the other three peptides, displaying a shoulder at 1628 cm$^{-1}$ and a maximum at 1645 cm$^{-1}$, indicating a combination of β-sheet and random-coil conformations[36,54], likely due to the diphenylalanine motif known to contribute to formation of β-sheet assemblies[55]. Interestingly, *α3-scrambled* generated a maximum at 1627 cm$^{-1}$ and a shoulder at 1684 cm$^{-1}$, indicating an antiparallel β-sheet conformation. Secondary structure prediction using Jpred[56] indicated an extended β-chain of *α3-scrambled*, in contrast to the α−helical pattern predicted for the native sequence. Like native PSMα3, *α3-EG/K*, in which the lysine residues were capped, formed α-helical structure, also reflecting similar amphiphilic pattern of hydrophobic and hydrophilic residues, accounting for the hydrophilicity of the ethylene glycol capped lysine residues.

Figure 3d depicts the CD spectra of the PSMα3 variants. The CD trace of native PSMα3 indicates a typical α-helical structure (blue spectrum), with two minima at 210 nm and 225 nm. *α3-cationic* generated low-intensity spectrum indicating that most of the peptide

aggregates thereby inducing light scattering (i.e., the micrometer-sized sheets in the cryo-TEM image, Fig. 3b). *α3-F10P* appear to have formed polyproline type II structure with a minimum at 198–200 nm, reflecting an unordered structure rich in proline residues, adopting turn-like rather than α-helical structure (confirmed also in the amorphous aggregates seen in the cryo-TEM analysis, Fig. 3b). *α3-(d)FF* yielded a typical β-sheet spectrum with minimum around 218 nm, corroborating the FTIR spectrum (-1630 cm$^{-1}$ shoulder, Fig. 3c) and the apparent fibrillar morphology seen in the cryo-TEM analysis. *α3-scrambled* generated a very low-intensity spectrum, likely linked to aggregation of the peptide. Figure 3d also shows that *α3-EG/K* produced a α-helical spectrum akin to the parent PSMα3.

Figure 3e, f portrays the PSMα3 variant-mediated catalytic nitrocefin amide bond hydrolysis, demonstrating pronounced divergence in catalytic performance. Figure 3e depicts the nitrocefin degradation curves recorded in the presence of PSMα3 variants at a concentration of 170 μM (prepared similarly to the experiment in Fig. 2c−incubated in water for two hours and buffered prior to

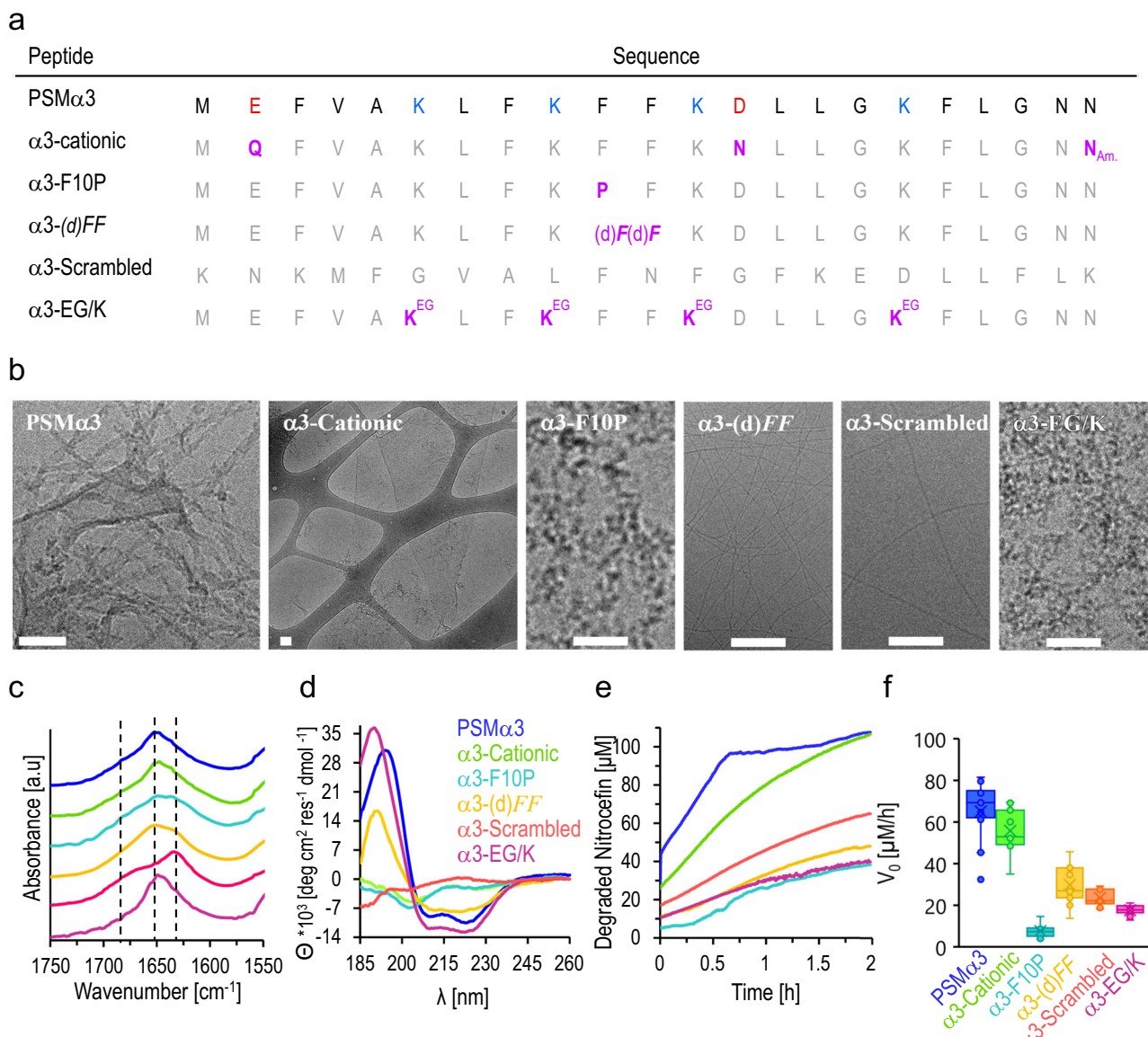

**Fig. 3 | Structural features and catalytic activities of PSMα3 variants.**
**a** Sequences of the PSMα3 variants. The modified residues in the variants are marked purple. *α3-cationic*: all acidic residues and the Asn-22 in the C-terminus were amidated. *α3-F10P*: Phe-10 was replaced with Pro residue. *α3-(d)FF*: two D-Phe residues instead of the L-handed residues. *α3-EG/K*: amine sidechains in the Lys residues were capped with two units of ethylene-glycol. **b** Cryo-TEM images of the PSMα3 variants (concentrations of 300 μM). Bars correspond to 100 nm. The experiment was repeated three times with independent peptide samples. **c** Amide region FTIR spectra (vertical broken lines represent the values of 1684 cm$^{-1}$,

1650 cm$^{-1}$, and 1628 cm$^{-1}$). **d** CD spectra of the PSMα3 variants. **e** Nitrocefin degradation reaction kinetics in the presence of PSMα3 variant assemblies formed after 2-h incubation in DIW and addition of Hepes buffer (peptide concentrations were 170 μM, initial nitrocefin concentration 107 μM). **f** Initial degradation rates, $V_O$, at initial nitrocefin concentration of 107 μM. Data presented in box-and-whisker plot, with mean line and quartile calculation using inclusive median, $N_{PSMα3} = 14$, $N_{α3\text{-}cationic} = 9$, $N_{α3\text{-}cationic} = 8$, $N_{α3\text{-}(d)FF} = 9$, $N_{α3\text{-}scrambles} = 8$, $N_{α3\text{-}EG/K} = 9$. The same color-coding was used in panels c-f. Source data are provided as a Source Data file.

mixing with nitrocefin). PSMα3 exhibited the most significant activity, degrading the entire nitrocefin reservoir (107 μM) after ~40 min, thereby reaching a plateau. *α3-cationic* exhibited similarly significant catalytic activity as native PSMα3, although at seemingly slower rate and without reaching an early plateau, likely reflecting the abundant amine residues at the amyloid surface. Notably, all other PSMα3 variants displayed lower catalytic performance (*α3-scrambled* featured somewhat more pronounced catalytic activity, red curve in Figs. 3e and S7).

The bar diagram in Fig. 3f further illuminates the catalytic profiles of the PSMα3 variants, outlining the initial rate of nitrocefin degradation reaction ($V_O$), calculated from the kinetic curves in Fig. 3e, f demonstrates that native PSMα3 and the *α3-cationic* variant exhibited

the highest initial reaction rates (around 60 μM/h), while the β-sheet-forming variants gave rise to lower initial rates (*α3-(d)FF* exhibited 30 μM/h, while *α3-scrambled*−21 μM/h) and *α3-EG/K* also exhibited low $V_O$ of 23 μM/h. *α3-F10P*, in comparison, had a very low catalytic activity characterized by initial reaction rate that was lower than 15 μM/h. Variant analysis was carried out in the case of PSMα2 (Fig. S8), furnishing similar structure/function insight on the catalytic activities of the peptide.

While Fig. 3 underlines the importance of the cross-α organization to the catalytic properties of PSMα3, we further analyzed the contribution of individual lysine residues to the catalytic properties. Lysines were shown to play a fundamental role in PSMα assembly, and lysine residues are believed to affect peptide virulence[38,57]. Figure 4a

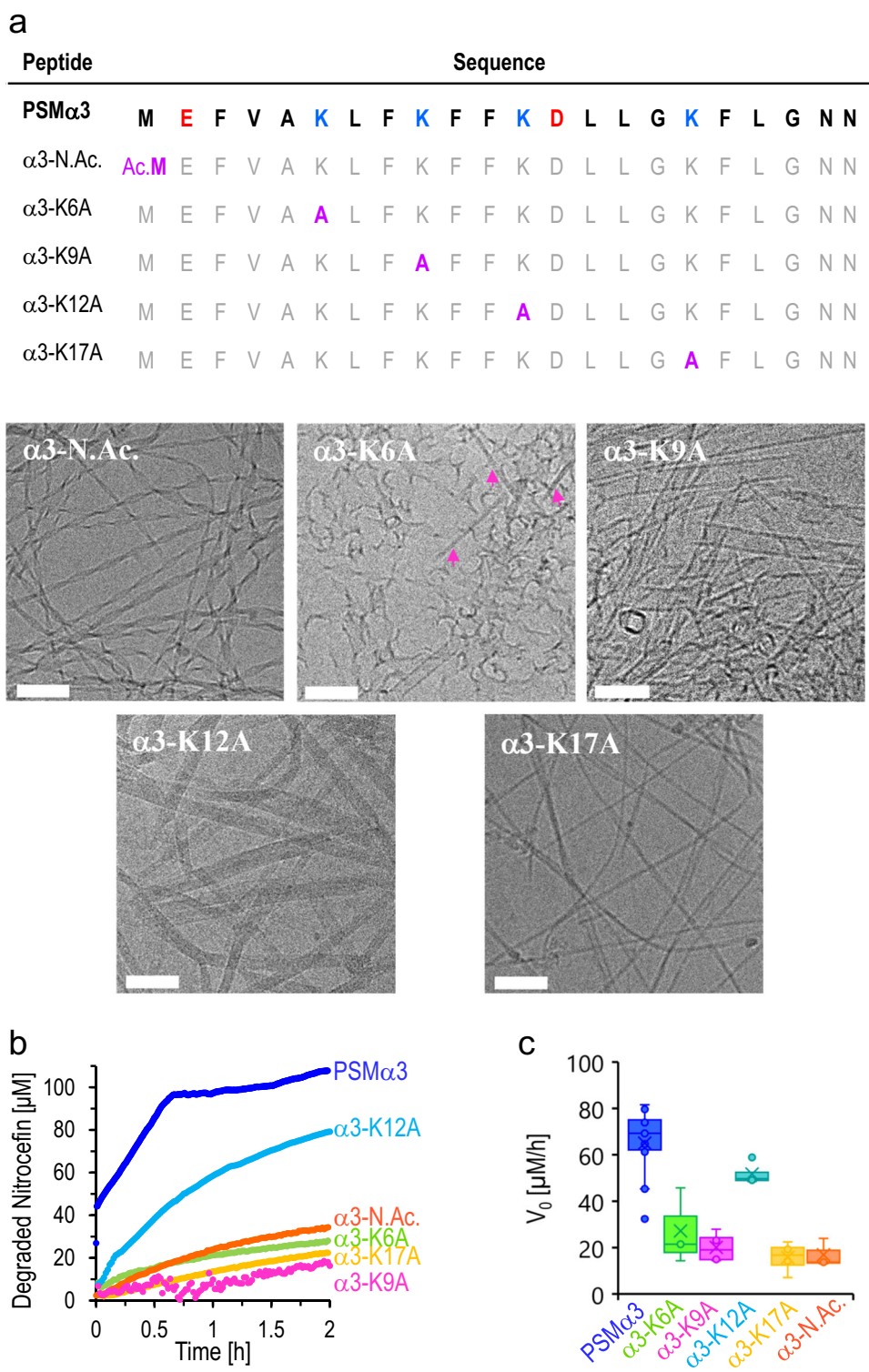

**Fig. 4 | Catalytic activities of lysine- and N-terminus-substituted PSMα3 variant fibrils. a** Sequences of the PSMα3 variants. The modified residues in the variants are marked purple. Cryo-TEM images of the PSMα3 variant assemblies (concentrations 300 µM; 2-h incubation in DIW and addition of Hepes buffer). Bars correspond to 100 nm. The experiment was repeated two independent peptide samples that were prepared separately. **b** Nitrocefin degradation kinetics in the presence of PSMα3 variant assemblies (peptide concentrations 170 µM, initial nitrocefin concentration 107 µM). **c** Initial degradation rates, $V_0$, at initial nitrocefin concentrations of 107 µM. Data presented in box-and-whisker plot, with mean line and quartile calculation using inclusive median, $N_{PSMα3} = 14$, $N_{α3\text{-}K6A} = 3$, $N_{α3\text{-}K9A} = 4$, $N_{α3K12A} = 4$, $N_{α3K17A} = 4$, $N_{α3\text{-}N.Ac} = 3$. The same color-coding was used in (**b**) and (**c**). Source data is provided as a Source Data file.

depicts point mutations targeting the primary amines in PSMα3, both alanine substitutions in each of the four lysine residues, as well as a PSMα3 variant in which the N-terminus was acetylated (denoted *α3-N.Ac*). The cryo-TEM images of the variants attest to the formation of fibrillar structures in all variants, albeit exhibiting distinct overall morphologies (Fig. 4a). Indeed, FTIR and CD spectroscopy data indicate that the PSMα3 variant fibrils adopted similar α-helical organization as native PSMα3 (Fig. S9).

Figure 4b, c characterize the catalytic nitrocefin hydrolysis, induced by the amine-substituted mutants. In general, all amine-substitutions led to a decrease in the catalytic activity in comparison to the parent PSMα3. Specifically, α3-N.Ac., α3-K6A, α3-K9A, and α3-K17A displayed minimal degradation of nitrocefin after two hours, reflected in the low concentration of the degradation product and low $V_0$ values. These findings indicate that the primary amines of the N-terminus and the side chains of K6, K9, and K17 play critical roles in the catalytic activity of PSMα3 fibrils. Interestingly, α3-K12A fibrils exhibited more pronounced catalytic activity albeit lower than the parent peptide, generating 70% substrate degradation after two hours and indicating that K12 differs in its contribution to the catalytic active site and/or the reaction mechanism. pH titration experiments (Fig. S10) reveal that the pKa values of the amine moieties in the PSMα3 fibrils were close to physiological pH (7.4), indicating ready occurrence of deprotonation, thereby enhancing the nucleophilic properties of the lysine residues (Fig. S10).

Molecular dynamics (MD) simulations were carried out, designed to probe the nitrocefin binding modality onto the surface of PSMα3 fibrils, and furnishing a comprehensive overview of the structural and mechanistic features of the catalytic process (Fig. 5). To facilitate the MD analysis, we constructed a minimal representation of the amyloid

surface (Fig. 5a) and placed nitrocefin in an unbound state approximately 2 nm from the surface (Fig. S11). We then aggregated 100 μs of simulation time over 20 repeat simulations of the binding process obtaining extensive sampling of ligand adsorption, binding conformations, and transitions between them. Binding site clustering using the PyLipID software[58] reveals two distinct bound conformations (Fig. 5bi,ii) based on the longest average binding duration of nitrocefin, which is ~300 ns for both sites. Inspection of the bound conformations indicates that distinct structural features are shared between the conformations. Specifically, the two bound nitrocefin conformations are anchored in the cavity between two PSMα3 helices by the thiophene moiety, which binds in a hydrophobic pocket consisting of phenylalanine F8 and F10. Likewise, the dinitrostyryl moiety binds in the F8, F10 pocket in both conformations, but in the adjacent cavity to the thiophene. Interestingly, the conformations mainly differ in the orientation of the β-lactam carbonyl of the 4-membered ring that is susceptible to be hydrolyzed resulting in nitrocefin degradation. In the first binding site the carbonyl binds the two central lysines, K9 and K12 (Fig. 5b,i), while in the second site, the carbonyl binds to the N-terminal primary amine of methionine M1 and lysine K6 (Fig. 5b,ii). Given the very slow turnover, covalent trapping of β-lactam intermediates may occur, accounting for the nucleophilic properties of the lysines.

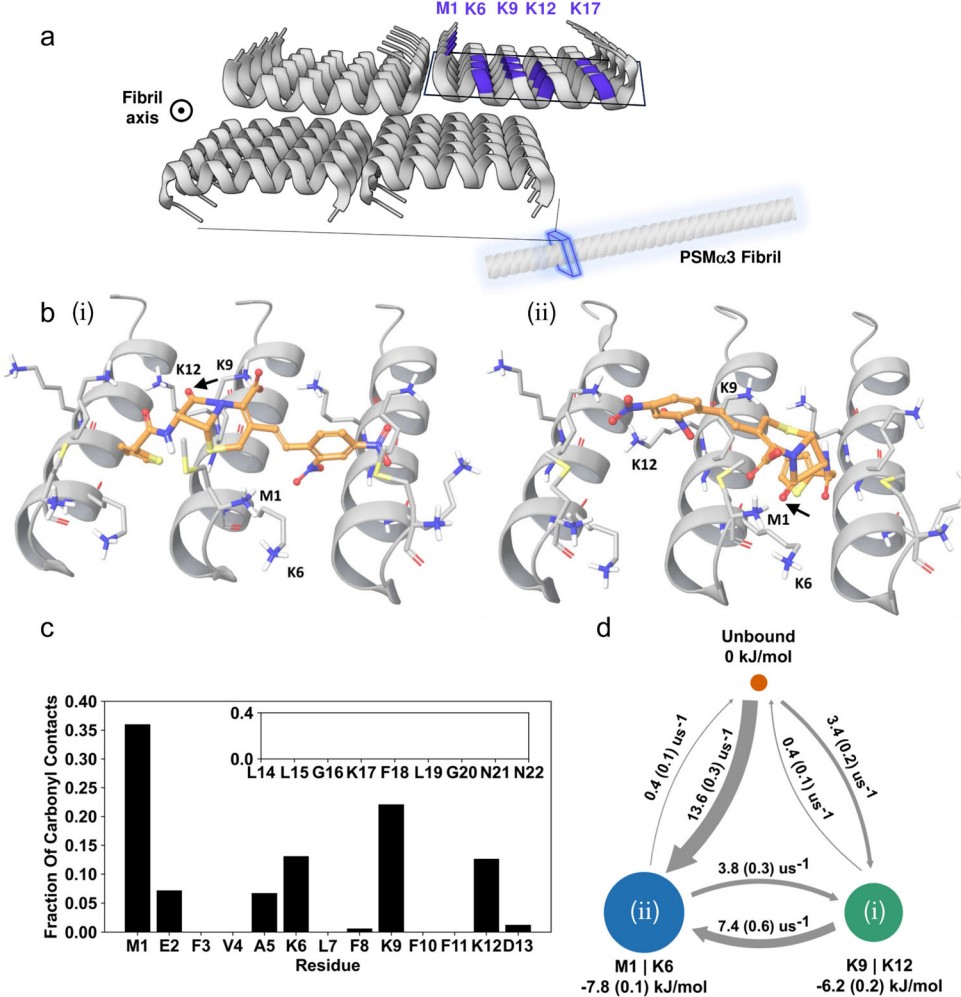

**Fig. 5 | Molecular Dynamics (MD) simulations of nitrocefin binding to PSMα3 amyloid fibrils' surface. a** The cross-α amyloid fibrils of PSMα3, composed of bilayer made of amphiphilic α-helices, display arrays of lysine residues (marked in purple, based on PDB structure 5I55[23]). **b** MD simulations reveal two distinct binding conformations of nitrocefin to the fibril surface (conformation (i) and (ii)). In both conformations, the β-lactam ring binds the primary amines of PSMα3 (β-lactam carbonyl marked with a black arrow). **c** The fraction of contacts between the β-lactam carbonyl and the peptide residues, calculated with a 0.45 nm contact cutoff. **d** Three-state Bayesian Markov State Model between the unbound state and the bound states M1|K6 (**b**, right) and K9|K12 (**b**, left). The free energy for each state at 20 °C and transition rates between the states were calculated from the MSM transition matrix. The sample standard deviation is shown in parentheses.

Complementing the static observations illuminating nitrocefin binding to the PSMα3 fibrils, we evaluated the dynamical contacts between the β-lactam carbonyl to each residue over the trajectories (Fig. 5c). The bar diagram in Fig. 5c outlines the fractions of the contacts attributed to each residue, with contact being defined as the carbonyl oxygen distance to any atom in a given residue being under 0.45 nm, a common cutoff for contacts in atomistic MD simulations[59]. Strikingly, M1 (containing the positively charged N-terminal), K6, K9, and K12 account for over 80% of the carbonyl-residue contacts, consistent with the observations for the binding site clustering (Fig. 5b), while residues 14 to 22, including K17, display very few contacts to the β-lactam carbonyl (Fig. 5c, inset).

To elucidate the binding free energy and kinetic relationship between the two bound states (depicted in Fig. 5b) and unbound state of nitrocefin, we constructed a three-state Bayesian Markov State Model using our simulation trajectories (Fig. 5d, see Methods). Motivated by the contact fractions (Fig. 5c) we define two bound states where the 4-ring β-lactam carbonyl is either closest to M1 or K6, denoted M1 | K6, corresponding to state Fig. 5b(ii), and another state where the carbonyl is closest to K9 or K12, denoted K9 | K12 corresponding to Fig. 5b(i). We find the free energy of the two bound states to be similar at −7.8 kJ/mol for the M1 | K6 state and −6.2 kJ/mol for the K9|K12 state, though indicating that M1 | K6 is slightly more stable. The transition rate from the unbound state to the M1 | K6 state is considerably faster compared to the K9|K12 state, suggesting that M1 and K6 are important for the initial binding of nitrocefin to PSMα3. The Markov State Model also reveals that the two bound states are kinetically linked with an estimated 3.8 transition events per microsecond going from the M1 | K6 state to the K9 | K12 state. This is in line with visual inspection of the trajectories where the dinitrostyryl moiety can unbind and rebind to a F8, F10 cavity two fibril registers down while the thiophene moiety remains anchored. Taken together, the MD simulation data highlight the importance of primary amines in M1, K6, K9, and K12 in coordinating the β-lactam's carbonyl. Importantly, the kinetic and thermodynamic analyses in Fig. 5 are consistent with the catalytic results recorded for the lysine-to-alanine substitutions (Fig. 4).

We tested the generality of PSMα3 amyloid catalyzed antibiotic hydrolysis, demonstrating that PSMα3 fibrils degrade additional, clinically relevant β-lactams (Fig. 6). The bar diagrams in Fig. 6 depict the hydrolysis kinetics, obtained via liquid chromatography-mass spectrometry (LC-MS), of amoxycillin, a β-lactam antibiotic used for treatment of severe infections[60] and is found on the list of essential medicines of the World Health Organization[61] (Fig. 6a), and penicillin-G, one of the most widely used antibiotics[60] (Fig. 6b). Strikingly, the LC-MS data in Fig. 6 demonstrate that PSMα3 fibrils hydrolyzed both amoxycillin and penicillin-G to a significantly higher extent compared to the spontaneous degradation of these antibiotic compounds.

To complement the intriguing phenomenological and mechanistic analyses in Figs. 1–6, we further evaluated whether PSMα3-induced catalysis of antibiotic degradation can be discerned in actual biological scenarios (Fig. 7). Figure 7a depicts bacterial growth curves of Staphylococcus aureus (ATCC 25923), upon exposure to nitrocefin at a concentration of 100 μg/mL, with and without PSMα3 pre-formed fibrils (peptide concentration 100 μM) added to the growth medium. As expected, nitrocefin significantly reduced bacterial proliferation (red curve) due to its antibiotic action. Importantly, however, addition of PSMα3 amyloid fibrils to the bacterial growth medium blocked the antibiotic effect, resulting in a similar growth curve as the control bacterial suspension (blue curve), ascribed to likely degradation of the nitrocefin molecules and concomitant disruption of their cell killing capabilities. PSMα3 fibrils alone did not disrupt bacterial proliferation (green curve). Somewhat increased absorbance of the samples containing PSMα3 amyloid fibrils (Fig. 7a) may be attributed to enhanced turbidity due to electrostatic interactions between the cationic peptides and anionic polysaccharides on the bacterial surfaces and concomitant aggregate formation. Importantly, different than PSMα3, exposure of S. aureus to nitrocefin in the presence of PSMα1 or PSMα4 aggregates did not block the antibiotic effects of the β-lactam (Fig. S15).

Figure 7b depicts the catalytic hydrolysis of nitrocefin placed upon biofilms produced by S. aureus, Salmonella enterica, and Escherichia Coli. In the experiments, the biofilms were generated by the bacteria upon 48-h incubation. Following washing and removal of unbound soluble bacteria, fresh nitrocefin solution (100 μg/mL, Hepes 50 mM) was placed upon the biofilms and the absorbance (480 nm) of degraded nitrocefin was measured after five-hour incubation (as control, self-degradation of nitrocefin was recorded). Remarkably, the S. aureus biofilm gave rise to considerable nitrocefin hydrolysis (absorbance around 1.0, Fig. 7b), significantly greater than the corresponding biofilms of either S. enterica or E. coli. This result may reflect the secretion of abundant PSMα fibrils in S. aureus biofilms[62], and absence of these peptides in biofilms of the two Gram-negative strains tested.

## Discussion

While PSMα's secreted by S. aureus have been reported as framework constituents in biofilm scaffoldings and as toxic species to other microbes or host cells, the functions of PSMα amyloids are still not fully resolved. Here, we demonstrate for the first time that native PSMα amyloids catalyze degradation of the prominent antibiotic β-lactam. Interestingly, PSMα2 and PSMα3 featured the highest catalytic activity towards amide bond hydrolysis of nitrocefin, compared to the two other sequences tested, PSMα1 and PSMα4. Indeed, while PSMα1 and PSMα4 were shown to contribute to S. aureus biofilm integrity and mechanical stability[63], PSMα3 was also reported to play a role in inflammatory processes, virulence and cytolytic activity of methicillin-resistant S. aureus (MRSA)[38]. PSMα3-mediated catalytic degradation of antibiotic compounds may constitute an important additional factor in S. aureus virulence and antibiotic resistance.

The divergent structural and catalytic properties of the PSMα3 variants (Figs. 3, 4) underscore important structure/function relationships pertaining to the catalytic nitrocefin degradation process. PSMα3 amyloids exhibit core structural features which may contribute to nitrocefin catalysis: their cationic nature arising from the four surface-exposed lysine residues, and the helical amphiphilic pattern[50,62,64]. Despite these structural elements, amyloids in general and PSMα aggregates in particular tend to exhibit polymorphism[46,65,66]; PSMα3 specifically was shown to adopt tubular and fibrillar aggregates in the same samples[67]. Notably, the ordered lysine arrays and the N-terminus generate higher local positive charge thus enabling enhanced adsorption of anionic molecules and increased reactivity via nucleophilic attack.

The structure/function analyses in Figs. 1–5 illuminate the mechanistic features of PSMα amyloid fibril catalyzed β-lactam hydrolysis, underscoring the key roles of the cross-α amyloid structure and the contributions of the lysine residues. Specifically, PSMα3 variants not displaying α-helical organization exhibited lower activity (even when fibrillation was preserved, Fig. 3). Helical peptide motifs provide softer and less rigid environment in comparison to β-sheet[68] possibly contributing to the flexibility of the active site upon substrate binding. Moreover, cross-α structures may provide higher surface area and exposed amphiphilic niches (as seen in the MD simulation, Fig. 5) functioning as active sites. Indeed, short synthetic cross-α peptides were recently shown to catalyze hydrolysis reactions while exposing seryl-histidine group on the fibril surface[69]. Furthermore, the cross-α PSMα3 fibrillar organization (consisting of two peptides in each layer[70]) effectively buries the hydrophobic residues while the hydrophilic residues are exposed on the fibril surface[23].

The PSMα3 variant data and MD simulations indicate that the primary amines (lysine side chains and N-terminus) play dual roles,

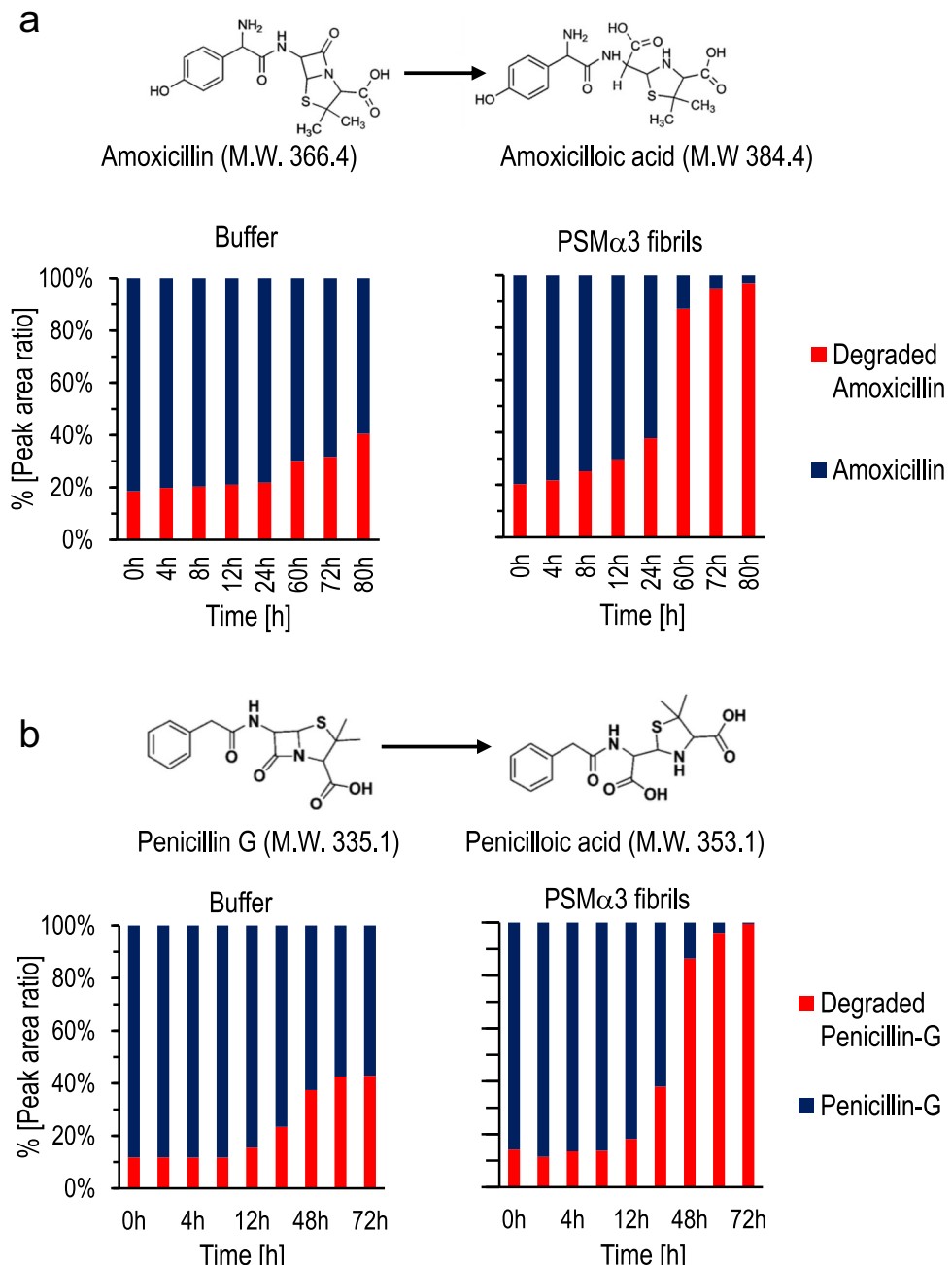

**Fig. 6 | Catalytic hydrolysis of clinical β-lactams by PSMα3 amyloid fibrils.** Degradation of amoxicillin (**a**) and penicillin G (**b**) was determined over time by LC-MS. In the experiments, mixtures of the antibiotic compounds (40 μM), and buffer (control) or PSMα3 fibrils (308 μM) were incubated at the indicated times. In the specific timepoints, the reaction was stopped upon addition of an organic solvent (acetonitrile in the case of amoxicillin, methanol for penicillin-G). The indicated values represent the percentages of the LC peaks corresponding to the parent compound and hydrolyzed product, respectively. Source data are provided as a Source Data file.

both serving as anchors for the negatively charged nitrocefin, and potentially acting as nucleophiles (via the amine side chains[32]) in their deprotonated state. Likewise, the primary amines can stabilize putative polyanion transition states. Indeed, PSMα2 and PSMα3, which sequences contain four lysines, exhibited the most pronounced catalytic activities, in comparison with PSMα1 and PSMα4, both peptides with three lysine residues. De-novo designed amphiphilic β-sheet amyloidogenic peptides decorated with lysines exhibited catalytic activity towards enantioselective chemical reactions[34,35]. Lysine residues in PSMα peptides have also been shown to play roles in receptor-mediated virulence of *S. aureus*[38].

The alanine screening catalytic analyses reveal that the N-terminal amine of PSMα3 and Lys residues in positions 6, 9, and 17 play key roles in nitrocefin hydrolysis. Interestingly, although Lys-17 is present in the catalytically active PSMα2 and PSMα3 (and not in PSMα1 and PSMα4), MD simulations predict negligible nitrocefin-binding to Lys-17 in PSMα3 (Fig. 5c, inset). These observations suggest that Lys-17 does not participate in the initial fibril-substrate complexation, but rather acts as a nucleophile in the catalytic process. Indeed, the pKa values of the primary amines (which are significantly lower than the free amino acids, Fig. S10) were close to the experimental (physiological) conditions, making it likely that some of the amines on the PSMα3 fibrils' surface are deprotonated, acting as nucleophiles (similar effects were

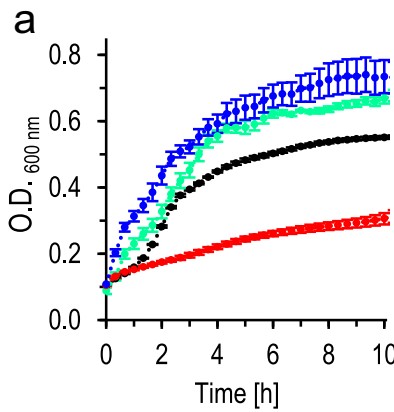

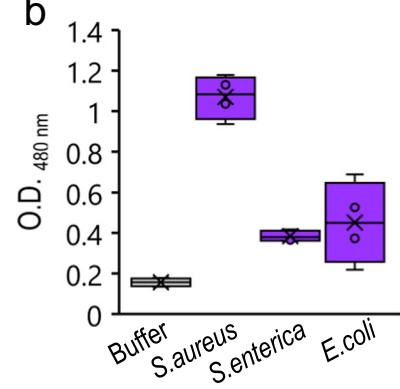

**Fig. 7 | Effects of PSMα3 amyloid fibrils upon nitrocefin exposure in physiological scenarios. a** Growth curve of *S. aureus* (black), *S. aureus* in the presence of nitrocefin (concentration of 100 μg/mL; red), co-addition of preformed PSMα3 fibrils (green), and in the presence of both PSMα3 fibrils and nitrocefin (blue). PSMα3 was preincubated in DIW and added to the growth media at a final concentration of 100 μM prior to addition of nitrocefin. Results are presented as an average ± SEM, *N* = 3 of independent bacteria samples that were prepared with/ without independent PSMα, treated and analyzed separately. **b** Absorbance of degraded nitrocefin at 480 nm following incubation of nitrocefin (100 μg/mL) with washed biofilms produced by *S. aureus, Salmonella enterica*, or *Escherichia coli* (300 min incubation times). Results are presented in box-and-whisker plot, with mean line and quartile calculation using inclusive median, *N* = 4 of independent samples of biofilms that were grown independently and treated in parallel. Source data are provided as a Source Data file.

reported for β-lactamase enzymes[71]). Lys-17 may play an additional important role in stabilizing a polyanion transition state of the degraded β-lactam.

Importantly, PSMα3-mediated catalytic hydrolysis was manifested also for clinically relevant β-lactams (Fig. 6) and in physiological scenarios (Fig. 7), attesting to the generality of the phenomenon we report and its physiological relevance. Notably, while synthetic peptide assemblies were shown to exhibit catalytic activities, up to now no peptide catalyst was reported to induce β-lactam hydrolysis. Furthermore, while the catalytic efficiencies reported here are markedly lower than the corresponding rates in β-lactamases[72], they are more pronounced than catalytic processes in peptide aggregates hydrolyzing ester bonds in model substrates[26–28,36,37]. In general, it is expected that the catalytic activity of the more promiscuous PSMα3 amyloid fibrils may be significantly lower than β-lactamase enzymes, likely accounting for the lesser degree of 'lock-key' specificity between the substrates and amyloid fibrils' surface, and less defined binding pocket on the fibril surface in comparison with classical enzymes. The relatively low catalytic activity of PSMα3 amyloid raises the question of whether β-lactam hydrolysis occurs in physiological contexts. Notably, it is conceivable that over longer timespans of antibiotics exposure to bacterial biofilms (and their amyloid framework)–which is the usual circumstances in the "real world"–catalytic degradation of antibiotic compounds may become significant. Indeed, the pronounced hydrolysis of nitrocefin recorded in the presence of actual *S. aureus* biofilms (Fig. 7b) supports this scenario.

The observation that functional bacterial amyloids catalyze degradation of common antibiotic small molecules furnishes a mechanism for antibiotic resistance phenomena. While the main contribution of FuBAs was thought to be their presence in biofilm scaffolds thereby acting as physical barriers for shielding bacterial cells, catalytic degradation of antibiotic compounds may be a powerful mechanism for protecting bacteria. Previous studies have shown, for example, that FuBAs in biofilm constructs adsorb small molecule metabolites, aiding bacteria to survive in stress conditions[9]. Furthermore, different than other bacterial resistance mechanisms, the catalytic activities reported here are general and pertain to various β-lactams. Indeed, different than β-lactamases that are generally substrate-specific, functional amyloid catalysis may be pertinent for a wide range of β-lactam antibiotics. An intriguing possibility is that functional bacterial amyloids may have evolved prior to β-lactamases, acting as

primitive enzymes. Overall, this work may underscore a mechanistic pathways accounting for the growing threat of MRSA antibiotic resistance, particularly in the case of β-lactam antibiotics.

This study presents the evidence of catalytic properties of functional bacterial amyloids in general, particularly cross-α amyloids secreted by *Staphylococcus aureus*. We show that PSMα assemblies, mainly PSMα2 and PSMα3, catalyze amide-bond hydrolysis of β-lactam antibiotics. Comprehensive structure/function screening of PSMα3 fibril variants complemented by MD simulations illuminate key features of the catalytically active sites on the amyloid fibrils' surface. In particular, the experiments highlight the contribution of the cross-α peptide organization, the roles of lysine residues in anchoring the β-lactam molecules via electrostatic attraction onto PSMα3 amyloid fibril surface, and the nucleophilic activities of the amine sidechains resulting in hydrolysis of amide bond in the four-member β-lactam rings. PSMα3 fibril mediated hydrolysis was observed for several clinically relevant β-lactam antibiotics, and in physiological bacterial settings, underscoring the broader scope of the concept. In conclusion, our observations reveal a previously unknown function of bacterial amyloids and may point to major contributions of bacterial biofilms to antibiotic resistance.

## Methods
### Materials
Phenol Soluble Modulin α (PSMα) peptides and their derivatives were synthesized, purified by high-performance liquid chromatography to >95%, characterized by mass spectroscopy, and supplied as a lyophilized powder by GenScript (Piscataway, NJ).

4-(2-hydroxyethyl)−1-piperazineethanesulfonic acid) (Hepes, >99% purity) and Nitrocefin (3-(2,4-Dinitrostyryl)-(6R, 7R)−7-(2-thienylacetamido)-ceph-3-em-4-carboxylic Acid, M.W of 516.50) were purchased from Holland-Moran (Yehud, Israel). Ammonium-acetate (99% purity), 1,1,1,3,3,3-Hexafluoro-2-propanol (HFIP, 99% purity), were purchased from Sigma-Aldrich (Rehovot, Israel). Acetonitrile (LC-MS grade, 99.9% purity) was purchased from Beith Dekel (Raanana, Israel. Manufactured by JTBaker). Amytracker-680 fluorescent dye was supplied by Ebba biotech (Stockholm, Sweden).

**Peptide sample preparation.** PSMα peptides and derivatives were dissolved in HFIP and titrated with NH$_4$OH (30%) until the solution was clear and no turbidity was detected, to prevent aggregation and

solubilize in monomeric forms. A stock solution at a concentration of 10 mM was prepared and stored at −20 °C until use. For each experiment, the solution was thawed, and the required amount was separated to aliquots in glass test-tubes and dried by evaporation (0–40 mBarr) for two hours to remove the HFIP (in most cases, 20 μL of concentrated peptide was placed in each aliquot test tube). The dried peptide samples were dissolved in doubly ionized water (DIW) applying short vortex and incubated for two hours at room temperature at peptide concentration of 444 μM. Then, buffer and DIW were added to dilute the sample to final concentration of 400 μM (in all kinetic experiments the buffer was Hepes 50 mM, pH = 7.4, in CD experiments the Hepes concentration was 5 mM). The buffers were prepared with DIW, 18.2 MΩ cm (Barnstead Smart2Pure, Thermo Scientific).

**Amyloid quantification and labeling.** PSMα assemblies were prepared as described above, incubated in DIW for two hours at concentration of 440 μM and then buffered with Hepes(final PSMα concentration 400 μM, Hepes 50 mM). Amytracker-680 was diluted with DIW by 50. 55 μL of PSMα solution was mixed with 5 μL of Amytracker-680 stock solution (Amytracker-680 final dilution was *500). Then, the samples were placed in 384 well black plate, incubated for 10 min and then fluorescent measurement was applied (Excitation at 550 nm, emission 650 nm) at Biotek Synergy H1 plate reader (Biotek, Winooski, VT, USA).

**Circular Dichroism (CD) spectroscopy.** CD spectra were recorded in a range of 185–260 nm on a Jasco J-715 spectropolarimeter (Tokyo, Japan) with 0.01 cm quartz cuvettes. All samples were prepared in diluted Hepes buffer of 5 mM to avoid buffer-dependent noise and recorded at final peptide concentration of 170 μM. CD spectra were recorded at 0.5 nm wavelength data pitch at 20 °C and represents an average of four scans. Background CD signals of the buffer recorded and subtracted from the corresponding spectra. The ellipticity ($\theta$) was normalized in accordance with the pathlength, the peptide concentration and number of residues (presented as $\theta_{MRE}$).

**Fourier Transform Infrared Spectroscopy (FTIR).** Peptide samples were prepared as described above, in ammonium acetate buffer (10 mM, pH = 7.35). Ammonium-acetate buffer was used due to its ability to partially evaporate along the lyophilization, allowing smaller buffer fingerprint and noise. Following the incubation, the samples were placed in plastic test-tubes and cryogenically frozen by dipping them in liquid nitrogen. Following this, the samples were lyophilized at −30 °C to preserve the secondary structure (LABCONCO-Triad, Labotal, Israel). FTIR spectra were recorded using NICOLET 6700 FTIR spectrometer with attenuated total reflectance (ATR) system equipped with diamond crystal and DTGS detector (Thermo Fischer Scientific, MA, USA). The FTIR spectra were acquired from 122 scans at 4 cm$^{-1}$ resolution and were corrected for spectral distortion using atmospheric suppression. A baseline correction function was applied to all spectra. Reference spectra were measured using bare ATR crystal surface.

**Cryogenic transmission electron microscopy (cryo-TEM).** a 5 μL droplet of each peptide's solution, prepared as reported above, was deposited on a glow-discharged TEM grid (300 mesh Cu Lacey substrate grid; Ted Pella). The excess liquid was automatically blotted out with a filter paper and the specimen was rapidly plunged into liquid ethane precooled with liquid nitrogen in a controlled environment (Leica EM GP). The vitrified samples were transferred to a cryo-specimen holder and examined at −181 °C with a FEI Talos S200C microscope operated at 200 kV in low-dose mode. Images were recorded using a Ceta camera (4k × 4k) and analyzed using Digital Micrograph Gatan Inc. software.

**Nitrocefin hydrolysis kinetics.** Nitrocefin, a β-lactam widely used as a surrogate substrate of Penicillin in β-lactamase activity assay, was used as a substrate. NC degradation was monitored for 2 h. Nitrocefin stock solution was dissolved in acetonitrile (20 mM) and then diluted with Hepes buffer (50 mM, pH 7.4) to nitrocefin concentration of 360 μM (further dilutions to lower concentrations were made with the 360 μM NC stock). All nitrocefin samples were kept on ice in cold conditions and any further dilution or mixing was performed in ice-cold conditions to avoid self-hydrolysis background reaction. 120 μL of the pre-cooled nitrocefin solutions were mixed inside precooled clear 96-well plate (Greiner F-bottom) with 60 μL of pre-incubated PSMα solution (510 μM PSMα, incubated in DIW and later buffered in Hepes 50 mM, pH 7.4). Immediately after mixing the samples, the plate was transferred to 20̊ °C pre-cooled plate reader (Multiskan GO, Thermo Scientific, Waltham, MA) and shaken slowly for 10 s. Then, the absorbance of degraded nitrocefin was measured for 2 h at interval of 20 s at $\lambda$ = 480 nm. Initial production rates ($V_0$) were calculated by slope of the initial linear range of degraded-nitrocefin production at the first 15 min. The reported results are and average of at least five separate measurements (the exact number of repeats, $N$, is reported next to the values). Each experiment also included blank samples of nitrocefin substrate diluted with buffer instead of PSMα. All presented results include subtraction of buffer control. The initial rate calculations were performed while also subtracting the self-degradation of the substrate, to ensure that this product absorbance is a result of PSMα presence.

**Kinetic data fit to the Michalis–Menten (MM) model.** The initial rates of degraded-nitrocefin production were calculated from the slopes of the first 15 min of measurement as reported above. The initial rates ($V_0$) were fitted using non-linear regression applied with the solver function in excel, either to Michalis–Menten equation

$$V = \frac{V_{max.}[S]}{(K_M + [S])}$$

Where $V_{max.}$ Stands for maximal reaction rate, $K_M$ stands for Michalis–Menten constant, [S] stands for the substrate molar concentration. Each initial rate value was presented as average ± SEM. The kinetic parameters were calculated by fitting the data to MM model using non-linear regression, and the confidence intervals for each parameter were calculated accordingly.

## Molecular dynamics simulations

**Simulation setup.** In order to simulate Nitrocefin binding to PSMα3, we create a minimal representation of the amyloid surface of a PSMα3 fibril. The system setup starts from the crystal structure of the PSMα3 fibril (PDB: 5I55), keeping only six copies of PSMα3 which are placed in the bottom of a box of size 5.8920 × 3.1530 × 4.6557 nm$^3$. The x-y box vectors are a multiple of the crystal unit cell, such that the system represents an infinite fibril surface of PSMα3 assuming periodic boundary conditions (Fig. S11). In this setup the hydrophobic side of the PSMα3 α-helix is exposed to the solvent and the protein backbone heavy atoms are therefore restrained with a harmonic potential of 1000 kJ/mol to keep the fibril intact. Nitrocefin is placed ~2 nm above the PSMα3 fibrils' surface in four distinct starting positions to limit bias from the starting configuration. A spike potential of 1000 kJ/mol was placed above Nitrocefin such that it cannot diffuse across the periodic boundary to the hydrophobic side of PSMα3. CHARMM36m force field parameters for the protein and CGenFF parameters for Nitrocefin were obtained from the CHARMM-GUI webserver[73–75]. Generic CHARMM36m simulation settings were also downloaded from CHARMM-GUI and used with only slight modifications, in short: All simulations were performed at 293.15 K and 1 bar in the NP$_z$T ensemble (0-compressibility in $x$–$y$ to keep the fibril unit cell constant) using the

Nose-Hoover thermostat and Parrinello-Rahman barostat. Hydrogen bonds were constrained using LINCS[76] and the equations of motion were integrated using a 2 fs timestep. The four systems were minimized and equilibrated for 1 ns, followed by five production simulation repeats (re-sampling velocities) of 5 µs resulting a combined 100 µs of simulation time. Frames were saved every 500 ps.

**Simulation Analysis.** To analyze the binding conformations of nitrocefin we use the PyLipID v1.5.14[58] python package to first cluster the protein residues which bind simultaneously to Nitrocefin into binding sites using the PyLipID function *compute binding nodes*. We report the top ranked ligand conformation for the top two binding site clusters based on longest average binding duration to the site using the function *analyze bound conformations*. To gain insight into the energetics and kinetics between the bound conformations, we create a 3-state reversible Bayesian Markov State Model[77,78] (MSM) based on the distance between the nitrocefin β-lactam carbonyl oxygen and each residue in PSMα3, motivated by the PyLipID determined conformations. We used the following rules to define the three states: Motivated by the contact fractions (Fig. 5c), we define the unbound state to be when the minimum distance from the β-lactam carbonyl to any residue is *not* to M1, K6, K9, K12; the M1|K6 state when the minimum distance is to either M1 or K6; and the M9|K12 state, when the minimum distance is to either M9 or K12. Bayesian MSMs differ from regular maximum likelihood MSMs in the estimation of the MSM transition matrix. In a Bayesian MSM the transition matrix is sampled from the posterior distribution[78] given the transition count matrix, allowing for uncertainty estimation. In this work, we use 10,000 resamples of the transition matrix to estimate the sample mean and standard deviation of the MSM derived properties. The MSM lag time was found by implied timescale analysis[78,79] (Fig. S12). We use the MSM to calculate inverse mean first passage times between states, corresponding to approximate diffusion transition rates between the states. The MSM stationary distributions were used to calculate the free energy of each state at 293.15 K. The unbound state was set to 0 kJ/mol. The MSM was created using the Python packages PyEMMA v2.5.11[80] and Deeptime v0.4.2[81].

**Liquid chromatography–mass spectrometry (LC-MS).** Amoxicillin or Penicillin solutions were dissolved in Hepes buffer (50 mM, pH 7.4) in concentration of 50 µM. PSMα3 solution in concentration of 1.54 mM was dissolved in DIW and Hepes buffer was added to the final concentration of 50 mM. The antibiotics' solutions and pre-incubated PSMα3 solutions were mixed to final PSMα3 concentration of 308 µM and antibiotic concentration of 40 µM. samples of 60 µL were collected in time intervals of 0, 4, 8, 12, 24, 60, and 72 h and immediately mixed with 180 µL organic solvent to stop the reaction (acetonitrile for Amoxicillin, methanol in the case of Penicillin). The samples were collected and injected sequentially into the LC-MS analysis.

Waters ACQUITY UPLC I-Class PLUS System coupled with high-resolution Thermo Orbitrap Exploris 240 mass-spectrometer were used for analysis of Amoxicillin and Penicillin-G over time. 1 µL of each sample was injected into the reverse-phase Acquity UPLC BEH C18 1.7 µm, 2.1 × 50 mm column (Waters) and run under a gradient program (Table S2 for Amoxicillin, Table S3 for Penicillin-G). The samples were ionized using soft electrospray ionization in positive mode and detected in a full scan mode in range from 60 to 900 m/z with a resolution of 240,000 Full Width at Half Maximum. MS was tuned according to the manufacture recommendations. Relative abundance of the intact/degraded compounds was calculated by peak area of the extracted ion chromatogram: for Amoxicillin 366.1–366.12, and degraded-Amoxicillin 384.1–384.13 m/z, Penicillin-G 335.1–335.12, for degraded-Penicillin-G 353.1–353.12, for (Figs. S13 and S14), respectively.

**In vitro investigation of *Staphylococcus aureus* sensitivity to nitrocefin in the presence of PSMα3.** The antimicrobial activity of nitrocefin against Staphylococcus aureus (ATCC 25923) was examined in the presence or absence of pre-assembled PSMα3, while monitoring the growth of the bacterial strain. PSMα3 fibrils were prepared as described above, in aqueous solution, in concentration of 100 µM. The bacterial growth was monitored in bacterial growth curves prepared as following: 3 µL of S. aureus microbial culture (OD = 1) were added to 100 µL LB media (Difco Luria-Bertani medium, BD, France). The bacterial solution was supplemented with 100 µM of the PSMα3 solution (dissolved and preincubated in DIW) or equivalent volume of DIW as a control. Then, aqueous solution of nitrocefin was added to each sample, generating final concentration of 100 µg/mL. As control, the growth curve of S. aureus was examined without any additives. All the samples were incubated while shaking in sterile clear 96-well plates at 37 °C for 16 h, and the absorbance at 600 nm was monitored every 20 min steps by microplate reader (Varioskan Flash, Thermo Scientific, Waltham, MA). Measurements were performed at three biological replicates, each with five technical replicates. The values were plotted in the graph as mean ± SD of representative results.

**Nitrocefin degradation by bacterial biofilms.** *Staphylococcus Aureus* (*S. aureus*, ATCC 25923), *E. Coli* K12 (ATCC 29425), and *Salmonella enterica* (*S. enterica*, ATCC 13076) strains were incubated for 16 h in LB broth medium (Lennox) at 37 °C. The bacterial suspensions were diluted 1:6 and incubated for ~3 h until the mid-log growth phase. After an additional dilution of 1:1000, 100 µL of the mid log growth suspensions were placed in each well of a 96-well plate (Thermo Fisher, Waltham, MA, USA). Afterwards, the plate statically incubated at 37 °C to allow biofilm formation. After 24 h, 100 µL of fresh LB broth was added to each well and the plate was incubated again at 37 °C for 24 h. After 48 h of incubation, all the planktonic bacteria were sucked out from each well and the biofilm that formed remained exposed on the surface of each well. Then, the preformed biofilms were incubated with 100 µL nitrocefin solution in concentration of 100 µg/mL, dissolved in PBS. Following incubation of 300 min the absorbance of the degradation product of nitrocefin was measured at 480 nm by microplate reader (Varioskan Flash, Thermo). As control, nitrocefin was dissolved in PBS at concentration of 100 µg/mL without the presence of bacteria or other biofilm components.

**Reporting summary**

Further information on research design is available in the Nature Portfolio Reporting Summary linked to this article.

## Data availability

Source data are provided.

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

## Acknowledgements

The measurements of electron microscopy, FTIR, and LC-MS were performed at the Ilse Katz Institute (IKI) for Nanoscale Science and Technology Ben-Gurion University of the Negev, Beer Sheva, Israel. The authors are thankful to Dr. Einat Nativ-Roth for help with the cryogenic TEM experiments. The authors are also grateful to Dr. Alexander Upcher for assistance with the microscopy measurements and Dr. Albert Batushansky from the metabolomics unit for LC-MS analysis. We are also grateful to Dr. Sofiya Kolusheva (BGU) and Prof. Daniel Otzen (Aarhus University) for their helpful discussions. E.A. is grateful for the support and funding of IKI, the European Molecular Biology Organization (EMBO) and Yad Hanadiv Foundation (the Rothschild Foundation). K.B.P. and B.S. acknowledge the Novo Nordisk Foundation ROBUST Grant NNF18OC0032608 for GPU computational resources used at the Centre for Scientific Computing Aarhus (CSCAA).

## Author contributions

E.A., K.B.P., M.L., and R.J. designed the research; P.A. performed and analyzed the pKa values of PSMα3; N.G. measured the cross-β bacterial amyloids activity; E.A. performed the experiments involving PSMα peptides and derivatives (CD spectroscopy, FTIR spectroscopy, nitrocefin degradation kinetics, TEM sample preparation, fluorescent labeling, and critical dissolution concentration); S.M.K contributed to the PSMα3 and derivatives measuring and performed the LC-MS measurements; K.B.P performed the molecular dynamics simulations; O.M performed all the in vitro experiments involving bacteria; H.R., B.S., M.L., and R.J. supervised the research; E.A., K.B.P., and R.J. wrote the manuscript. All authors have commented and approved the final version of the manuscript.

## Competing interests

The authors declare no competing interests.
