## [Peer Review File · Nature Communications]

Staphylococcus aureus functional amyloids catalyze degradation of β -lactam antibioticsREVIEWER COMMENTS

Reviewer #1 (Remarks to the Author):

This study uncovers a potential beta-lactamase activity for the helical phenol-soluble modulins (PSMs) from *Staphylococcus Aureus*. Using a classical spectrophotometric assay, the PSM α family, particularly PSM α 3, induces a measurable catalytic degradation of the antibiotic nitrocefin. The experiments have been carried out carefully and the catalytic efficiency reported accurately. They conclude that specific arrays of nucleophilic lysine residues on the surface of alpha-helical amyloid is key to the enzymatic activity. This would be a significant discovery is proven to play a role in bacterial antibiotic resistance, however the manuscript does not address this sufficiently.

The catalytic efficiency of nitrocefin degradation appears low compared to other bacterial beta-lactamase enzymes, but the manuscript does not discuss this at any length. While bacterial beta-lactamase enzymes have some degree of specificity they are generally more efficient than PSMs over a range of beta-lactams. The author state that this is a non-specific antibiotic mechanism which sets it aside from specific beta-lactamase enzymes. If so do PSMs break down other antibiotics and does the catalytic efficiency vary? This needs to be demonstrated.

How does this activity play a role in bacterial resistance to antibiotics in the wild when specific beta-lactamase enzymes and penicillin-binding proteins exist in *Staphylococci*? I would presume that if this is a genuine mechanism for antibiotic resistance in bacteria the lysine array in PSMs would have evolved with the rise in antibiotic usage – is this the case?

The manuscript states that cross beta amyloids are even less efficient than the cross alpha amyloid at degrading nitrocefin but they demonstrate this on one amyloid - Amyloid-beta, which is not a bacterial functional amyloid. The authors should include experiments with a bacterial cross beta-amyloid, such as curli, tas etc in which antibiotic degradation might be relevant.

The authors analyse the structural-function properties of PSM α 3 variants, the key results were obtained from peptides in which charged residues were chemically modified. Bearing in mind these are synthetic peptides this seems a rather clumsy approach. How about mutating individual residues to change the charge properties or structural arrangements of lysine residues on the amyloid surface. The study would benefit from a more site-directed approach.

How does the enzyme activity vary with pH – this might provide further evidence for their conclusions on the mechanism.

Reviewer #2 (Remarks to the Author):

PSMs – phenol soluble modulins – are short (~20 residues for the alpha variants) amphipathic peptide toxins secreted by specific ABC transporters in *S. aureus*. PSMs are able to assemble into amyloid structures with roles in biofilm formation and competition with other microbes or host cells. Here, Arad et al hypothesize that these amyloids also play a role in the catalytic degradation of beta-lactams. They compare different PSM α peptides showing specifically PSM α 3 and α 4 have the “highest” activity in hydrolyzing nitrocefin, a chromogenic beta-lactam typically used for beta-lactamase detection in vitro (it is not a clinical beta-lactam drug). They characterize various PSM fibers at very low resolution by cryoTEM which is not incredibly informative and discuss features of the predicted peptide amyloid structures that may correlate to hydrolytic activity. They conclude the more active ones have a higher amphipathic helical propensity (CD and FTIR) and increased occurrence of lysine residues, the latter implicated in the activity due to their complimentary charge with the anionic nitrocefin (a carboxylate functional group common to most beta-lactams).

This previously undescribed role of PSMs in bacterial defense is interesting and builds on the authors previous work on the enzymatic activity of other amyloids. While the results showing

enzymatic activity of PSM2a and 3a seem somewhat compelling, they are still very poor turnover rates, many orders of magnitude lower compared to for eg true beta-lactamases. Comparative discussion on these relative rates needs to be added. One wonders if these are indeed even physiologically helpful rates given fast diffusion/uptake of beta-lactam compounds into *S.aureus* cells for eg. Some information in this regard would be important to add context. Comparison of beta-lactam induced *S.aureus* killing with and without PSM expression and/or biofilm might help support. The kinetic measurements are fraught also by large errors and the apparent statement that nitrocefin was also undergoing spontaneous degradation (see below). The analysis of other substrates including clinical beta-lactams would have strengthened.

With the exception of a3-F10P, all the variant peptides show moderate initial degradation rates (Fig. 4F). Notably a3-(d)FF, which adopts a beta-sheet structure yet maintains activity ~50% of PSMa3, and EG/K, which still exhibits moderate activity ~30% that of wild-type in absence of any positive charge. The latter in particular calls in to question the conclusion that the activity is mediated by these lysine residues, which are proposed to play somewhat contradictory roles in both electrostatic attraction (requiring a full charge as expected of surface exposed lysines) and functioning as nucleophiles via the amine side chains (requiring presumably therefore a depressed pKa to be in the neutral nucleophilic form) (Page 17). In general, whilst there is a decrease in activity, it would seem substitution of these residues should be more significant if these are key catalytic residues. Further, how do the authors propose these lysines are selectively charged given their predicted surface location? If the amine is a nucleophile, is covalent catalysis proposed? What would stabilize the transition state? Protonate the leaving group? Or position a hydrolytic water? None of these questions are addressed here.

Overall, while the study in its current form is compelling in hinting at a novel role of PSMs and functional bacterial amyloids that would be of interest to a broad audience, the results provided do not suitably support the conclusions made.

Figures 1 and 5 are not informative and should be removed

Referring to a proposed catalytic mechanism (Figure 5) is misleading - this is not a flushed out mechanism being proposed, it is a clustering of residues from a modelled structure with two proposed roles (electrostatic binding of substrate, nucleophile, as above, many pieces of the mechanistic puzzle are not answered). It is carbonyl not the carboxyl that is the site for nucleophilic attack (mis named in the legend).

Methods – The authors make a statement that “The initial rate calculations were performed while also subtracting the self-degradation of the substrate, to ensure that this product absorbance is a result of PSM presence”. This implies the nitrocefin was undergoing spontaneous breakdown? This should be clearly described in the methods as to how this was quantitatively taken into account.

In the CD, some explanation as to the concentration used (peptides monomeric? Equilibrium of monomer and oligomer?) would have been useful.

In general, this paper needs careful editing, there are many awkward sentences and grammatical errors. Starting with the abstract, discussion of the catalysis and chemistry is often clumsy or incorrect “catalyze breakup of b-lactams” should be “catalyze hydrolysis of the b-lactam bond” or “catalyze hydrolysis of the amide-bond four membered ring” should be “catalyze hydrolysis of the amide like bond of the four membered b-lactam ring”

Reviewer #3 (Remarks to the Author):

Alpha phenol soluble modulins (PSMa) are functional amyloid expressed by *S. aureus*. The manuscript by Arad et al. characterizes the ability of aggregates formed by PSMa 1 to 4 peptides to hydrolyze nitrocefin, a β -lactam substrate. PSMa3 and PSMa2 catalyze nitrocefin hydrolysis better than PSMa1 and PSMa4. The authors identify that the lysine residues and cross- α structural

organization are important for the catalytic activity of PSMa3. The findings are interesting and nicely presented, but the study lacks important in vivo experiments with *S. aureus* in the presence of β -lactam antibiotics to support a biological role for the hydrolysis activity.

1. Are PSMa1-4 expressed by *S. aureus* in equal amounts? Do they assemble into hetero polymers in known ratios? Do PSM1-4 form homopolymers or heteropolymers on *S. aureus*? If *S. aureus* form heteropolymers can the authors comment on how that would impact the hydrolysis of β -lactams?

2. Mention the overall charge of the peptide in Figure 2A.

3. Does increasing the number of lysine residues in PSMa2 or PSMa3 increase their catalytic activity towards nitrocefin?

4. Polymorphism is common in protein aggregation. Is it known if the PSMs form polymorphic structures or do the aggregates formed by PMS peptides under different conditions have similar activity towards nitrocefin?

5. It is unclear if the authors tested a functional role for PSMs ability to hydrolyze β -lactams. Are *S. aureus* without PSMs more susceptible to β -lactam antibiotics? Have the authors tried any growth assay on *S. aureus* expressing each PSMa1-4 in the presence of antibiotics to validate the current finding.

July 29, 2023

Detained response to Reviewers' comments

Reviewer #1:

This study uncovers a potential beta-lactamase activity for the helical phenol-soluble modulins (PSM α s) from Staphylococcus Aureus. Using a classical spectrophotometric assay, the PSM α family, particularly PSM α 3, induces a measurable catalytic degradation of the antibiotic nitrocefin. The experiments have been carried out carefully and the catalytic activity reported accurately. They conclude that specific arrays of nucleophilic lysine residues on the surface of alpha-helical amyloid is key to the enzymatic activity. This would be a significant discovery is proven to play a role in bacterial antibiotic resistance, however the manuscript does not address this sufficiently.

We are grateful to the Reviewer for the thorough reading, favorable comments, and insightful, constructive comments. Based on the Reviewers' comments and suggestions we carried out additional experiments and revised the manuscript; below I address the specific comments of the Reviewer.

1. The catalytic efficiency of nitrocefin degradation appears low compared to other bacterial beta-lactamase enzymes, but the manuscript does not discuss this at any length. While bacterial beta-lactamase enzymes have some degree of specificity they are generally more efficient than PSMs over a range of beta-lactams. The author state that this is a non-specific antibiotic mechanism which sets it aside from specific beta-lactamase enzymes. If so do PSMs break down other antibiotics and does the catalytic efficiency vary? This needs to be demonstrated.

Response: We thank the Reviewer for this important comment. Indeed, as the Reviewer points out, we expect that the catalytic activity of the more promiscuous PSM α 3 amyloid fibrils will be lower than β -lactamases enzymes, primarily accounting for the lesser degree of "lock-key" interactions between the amyloid fibrils and substrates, and less defined substrate binding pocket on the fibril surface in comparison with classical enzymes. In that regard, while β -lactamases have evolved into more efficient enzyme adopting defined 3D structures of the active site, amyloids are simpler, albeit resilient assemblies, expected to be less catalytically efficient. Following the Reviewer's comment, we expanded the pertinent discussion of this aspect [page 29 lines 4-11].

To address the key issue of PSM α 3-mediated catalytic degradation of other β -lactams, we further tested additional compounds. Indeed, catalytic activity has been detected for penicillin-G and amoxicillin, well known and widely used antibiotics, underscoring the generic nature of the new phenomenon we are reporting. These data are presented in a new Figure 6, page 24.

2. How does this activity play a role in bacterial resistance to antibiotics in the wild when specific beta-lactamase enzymes and penicillin-binding proteins exist in Staphylococci? I would presume that if this is a genuine mechanism for antibiotic resistance in bacteria the lysine array in PSMs would have evolved with the rise in antibiotic usage – is this the case?

Response: We thank the Reviewer for this comment and interesting proposition. We are not aware in the literature of changes in the number of lysines in PSM α 3 due to evolutionary pressure. It should also be noted that positive charges partake in other activities of functional amyloids, such as cytolytic effects against host cells and other microbes. It is possible, however, that the ubiquitous use of antibiotics may have led, evolutionary speaking, to more pronounced *secretion* of PSM α 3, compared to other PSM α peptides containing fewer lysines. Indeed, the fact that different *Staph. aureus* strains produce different variety and abundance of PSMs (discussed in detail in pages 16, second paragraph and page 27, first paragraph (also seen in ref. 38 and 57) might be related to environmental stress including antibiotics.

3. *The manuscript states that cross beta amyloids are even less efficient than the cross alpha amyloid at degrading nitrocefin but they demonstrate this on one amyloid - Amyloid-beta, which is not a bacterial functional amyloid. The authors should include experiments with a bacterial cross beta-amyloid, such as curli, tas etc in which antibiotic degradation might be relevant.*

Response: We thank the Reviewer for this suggestion. To address this comment, we analyzed nitrocefin degradation by two well-known cross- β amyloids secreted by *Pseudomonas aeruginosa* – FapB and FapC (new Figure S3). The kinetic curves in Figure S3 reveal significantly less catalytic activities by these two fibril species, attesting to the prominent role of the cross-alpha amyloid organization.

4. *The authors analyse the structural-function properties of PSM α 3 variants, the key results were obtained from peptides in which charged residues were chemically modified. Bearing in mind these are synthetic peptides this seems a rather clumsy approach. How about mutating individual residues to change the charge properties or structural arrangements of lysine residues on the amyloid surface. The study would benefit from a more site-directed approach.*

Response: We are grateful to the Reviewer for this comment. Following this suggestion, we further measured the catalytic activity of additional PSM α 3 variants exhibiting point mutations in which the individual lysine residues within the sequence were substituted with alanine (i.e., K6A, K9A, K12A and K17A). An additional variant consisting of a protected (acetylated) amine at the N-terminus was also tested (noted α 3-N.Ac.). Indeed, the catalytic activities of these variants (presented in a revised Figure 5) shed light on the catalytic mechanism, specifically the key roles of K6, K9 and K12 in binding the carbonyl moiety of nitrocefin, particularly reflected in the comprehensive structural model in Figure 5.

5. *How does the enzyme activity vary with pH – this might provide further evidence for their conclusions on the mechanism.*

Response: We thank the Reviewer for this constructive suggestion. To address this issue and evaluate the effect of pH variation on the fibrils and their activities we measured PSM α 3 amyloid induced degradation of nitrocefin in three pH values (5, 7.4 and 9). These results are presented in a new Figure S10. The experiments indicate pK_a values of 7.8 and 8.1 for the amines, which are lower than the amines of free lysines, and may explain the deprotonation and enhanced reactivity of the lysine side chains.

Reviewer #2:

PSMs – phenol soluble modulins – are short (~20 residues for the alpha variants) amphipathic peptide toxins secreted by specific ABC transporters in S. aureus. PSMs are able to assemble into amyloid structures with roles in biofilm formation and competition with other microbes or host cells. Here, Arad et al hypothesize that these amyloids also play a role in the catalytic degradation of beta-lactams. They compare different PSMalpha peptides showing specifically PSMa3 and a4 have the “highest” activity in hydrolyzing nitrocefin, a chromogenic beta-lactam typically used for beta-lactamase detection in vitro (it is not a clinical beta-lactam drug). They characterize various PSM fibers at very low resolution by cryoTEM which is not incredibly informative and discuss features of the predicted peptide amyloid structures that may correlate to hydrolytic activity. They conclude the more active ones have a higher amphipathic helical propensity (CD and FTIR) and increased occurrence of lysine residues, the latter implicated in the activity due to their complimentary charge with the anionic nitrocefin (a carboxylate functional group common to most beta-lactams).

We thank the Reviewer for the thorough analysis of the manuscript and constructive comments and suggestions, which we address below.

1. This previously undescribed role of PSMs in bacterial defense is interesting and builds on the authors previous work on the enzymatic activity of other amyloids. While the results showing enzymatic activity of PSM2a and 3a seem somewhat compelling, they are still very poor turnover rates, many orders of magnitude lower compared to for eg true beta-lactamases. Comparative discussion on these relative rates needs to be added.

Response: As per our response to Reviewer # 1 (1st comment), we expect that the catalytic activity of the more promiscuous PSM α 3 amyloid fibrils will be lower than β -lactamases, primarily accounting for the lesser degree of “lock-key” interactions between the amyloid fibrils and substrates, and less defined substrate binding pocket on the fibril surface in comparison with classical enzymes. In that regard, while β -lactamases have evolved into more efficient enzymes adopting defined 3D structures of the active site, amyloids are simpler, albeit resilient assemblies, expected to be less catalytically efficient however exhibiting wider spectrum of substrate targets. Following the Reviewer’s comment, we expanded the pertinent discussion of this aspect [page 29, first paragraph].

2. One wonders if these are indeed even physiologically helpful rates given fast diffusion/uptake of beta-lactam compounds into S.aureus cells for eg. Some information in this regard would be important to add context. Comparison of beta-lactam induced S.aureus killing with and without PSM expression and/or biofilm might help support.

Response: This is indeed an important question. One has to consider that the bacterial cells are generally embedded within the biofilm matrix, and as such β -lactam antibiotics may still undergo amyloid-induced catalytic degradation prior to cell uptake, upon interactions and/or adsorption onto the biofilm matrix. Following the Reviewer’s suggestion, we demonstrated inhibition of *S. aureus* killing in the presence of PSM α 3 fibrils (Figure 7A); we also demonstrated antibiotic degradation induced by *S. aureus* biofilm (Figure 7B). Concerning engineered bacteria over-expressing PSMa3 – we are not aware of the availability of such strains, and this line of research is currently pursued in our laboratories.

3. The kinetic measurements are fraught also by large errors and the apparent statement that nitrocefin was also undergoing spontaneous degradation (see below).

Response: The nitrocefin spontaneous degradation is very low at neutral pH, and in any case was used as a blank in all parameter calculations.

4. *The analysis of other substrates including clinical beta-lactams would have strengthened.*

Response: we thank the Reviewer for this important suggestion; PSM α 3-catalyzed degradation of other β -lactams, particularly the clinically important penicillin and amoxicillin, has been observed and included in the revised manuscript (new Figure 6, page 24).

5. *With the exception of a3-F10P, all the variant peptides show moderate initial degradation rates (Fig. 4F). Notably a3-(d)FF, which adopts a beta-sheet structure yet maintains activity ~50% of PSM α 3, and EG/K, which still exhibits moderate activity ~30% that of wild-type in absence of any positive charge. The latter in particular calls in to question the conclusion that the activity is mediated by these lysine residues, which are proposed to play somewhat contradictory roles in both electrostatic attraction (requiring a full charge as expected of surface exposed lysines) and functioning as nucleophiles via the amine side chains (requiring presumably therefore a depressed pKa to be in the neutral nucleophilic form) (Page 17). In general, whilst there is a decrease in activity, it would seem substitution of these residues should be more significant if these are key catalytic residues. Further, how do the authors propose these lysines are selectively charged given their predicted surface location? If the amine is a nucleophile, is covalent catalysis proposed? What would stabilize the transition state? Protonate the leaving group? Or position a hydrolytic water? None of these questions are addressed here.*

Response: we are grateful to the Reviewer for pointing out these key mechanistic issues. Following this comment and to better characterize PSM α 3 fibrils surface properties and contributions to the catalytic mechanism, we evaluated the protonation conditions (Figure S10). The results attest that the pKa values of the lysine side chains are close to physiological pH. Therefore, local pH shifts and hydrogen-bonds with other residues on the fibril surface can affect the reactivity of the primary amines allowing covalent catalysis involving nucleophilic attack of the β -lactam carbonyl, followed by hydrolytic cleavage.

Molecular dynamics (MD) simulations modeling, included in the revised manuscript (new Figure 5), confirms the key roles of the interactions between the carbonyl moieties of nitrocefin and lysine residues and PSM α 3 N-terminus. This mechanistic model is further supported by screening of additional PSM α 3 variants in which individual Lys were substituted with Ala residues. Specifically, the MD simulations and K/A variant analyses point to the key roles of the N-terminal amine and K6, K9, K12 in substrate binding, and K-17 while not significantly involved in binding, intimately partakes in the catalytic reaction, probably acting as the main nucleophile. This conclusion is consistent with the observed catalytic activities of PSM α 3 and PSM α 2, which contain Lys in position 17, different than PSM α 1 and PSM α 4 which are much less catalytically active.

The pH-dependent nitrocefin hydrolysis results (new Figure S10) are also consistent with the occurrence of covalent binding between the fibril catalyst and the substrate leading to hydrolytic cleavage. Specifically, in pH 5, the final hydrolysis step is inhibited due to the low concentration of hydroxide residues in the aqueous solution as the fibril-substrate complex remains stable. In contrast, at pH 9, excess of hydroxide ions likely accelerates the enhanced decomposition of the fibril-substrate intermediate. A similar trend was reported in the case of hydrolytic reactions by synthetic catalytic peptides (for example, Díaz-Caballero et al. ACS catalysis 11.2 (2020): 595-607; Rufo et al. Nature chemistry 6.4 (2014): 303-309). Indeed, the refined catalytic mechanism

emerging from the mutant analyses and MD simulations echoes other hydrolytic processes catalyzed by synthetic β -sheet peptide (for example J. Am. Chem. Soc. 2020, 142, 9, 4098–4103).

Following the Reviewer's comments, a detailed discussion of the proposed catalytic mechanism, based on the PSM α 3 variant data and MD simulations, is included in the revised manuscript (page 27, last paragraph and page 29, first paragraph).

6. Overall, while the study in its current form is compelling in hinting at a novel role of PSMs and functional bacterial amyloids that would be of interest to a broad audience, the results provided do not suitably support the conclusions made.

Response: we are grateful for this assessment; as stated, a significant body of additional data has been included in the revised manuscript, supporting, in our opinion, the new observation reported and mechanistic analysis.

7. Figures 1 and 5 are not informative and should be removed

Referring to a proposed catalytic mechanism (Figure 5) is misleading - this is not a flushed out mechanism being proposed, it is a clustering of residues from a modelled structure with two proposed roles (electrostatic binding of substrate, nucleophile, as above, many pieces of the mechanistic puzzle are not answered). It is carbonyl not the carboxyl that is the site for nucleophilic attack (mis named in the legend).

Response: Following the Reviewer's suggestion we removed both Figures (a version of Figure 1 is included as panel 2A in the revised manuscript, designed to provide a visual guidance to the catalytic experiments carried out). Concerning the scheme depicting the proposed catalytic mechanism (originally in Figure 5), this is now substituted with a mechanistic model based upon the comprehensive MD analysis carried out (new Figure 4). Referral to carbonyl rather than carboxyl – corrected.

8. Methods – The authors make a statement that “The initial rate calculations were performed while also subtracting the self-degradation of the substrate, to ensure that this product absorbance is a result of PSM presence”. This implies the nitrocefin was undergoing spontaneous breakdown? This should be clearly described in the methods as to how this was quantitatively taken into account.

Response: The spontaneous degradation of nitrocefin at neutral pH at RT is very low, as seen in the blank sample at figure 2A. Yet, to keep the results as accurate as possible it is used as blank (with respect to the concentration of nitrocefin in the measured sample). Following the Reviewer's query and to better clarify this point, we added pertinent text in the Methods section (p...).

9. In the CD, some explanation as to the concentration used (peptides monomeric? Equilibrium of monomer and oligomer?) would have been useful.

Response: The CD spectroscopy was performed on pre-incubated and equilibrated PSM α peptides. Additional information was added in the Results section, page 16.

10. In general, this paper needs careful editing, there are many awkward sentences and grammatical errors. Starting with the abstract, discussion of the catalysis and chemistry is often clumsy or incorrect “catalyze breakup of b-lactams” should be “catalyze hydrolysis of the b-

lactam bond” or “catalyze hydrolysis of the amide-bond four membered ring” should be “catalyze hydrolysis of the amide like bond of the four membered b-lactam ring”
They characterize various PSM fibers at very low resolution by cryoTEM which is not incredibly informative and discuss features of the predicted peptide amyloid structures that may correlate to hydrolytic activity. They conclude the more active ones have a higher amphipathic helical propensity (CD and FTIR) and increased occurrence of lysine residues, the latter implicated in the activity due to their complimentary charge with the anionic nitrocefin (a carboxylate functional group common to most beta-lactams).

Response: we are grateful to the Reviewer for pointing out these important textual issues. The manuscript has been carefully rewritten and relevant terms corrected and revised.

Reviewer #3:

*Alpha phenol soluble modulins (PSM α) are functional amyloid expressed by *S. aureus*. The manuscript by Arad et al. characterizes the ability of aggregates formed by PSM α 1 to 4 peptides to hydrolyze nitrocefin, a β -lactam substrate. PSM α 3 and PSM α 2 catalyze nitrocefin hydrolysis better than PSM α 1 and PSM α 4. The authors identify that the lysine residues and cross- α structural organization are important for the catalytic activity of PSM α 3. The findings are interesting and nicely presented, but the study lacks important *in vivo* experiments with *S. aureus* in the presence of β -lactam antibiotics to support a biological role for the hydrolysis activity.*

Response: We thank the Reviewer for the favorable assessment and insightful comments. Following the Reviewer’s comment, we carried out an *in vivo* experiment demonstrating reduction of the antibiotic activity of nitrocefin towards *S. aureus* in the presence of PSM α 3 fibrils (Figure 7).

*1. Are PSM α 1-4 expressed by *S. aureus* in equal amounts? Do they assemble into hetero polymers in known ratios? Do PSM1-4 form homopolymers or heteropolymers on *S. aureus*? If *S. aureus* form heteropolymers can the authors comment on how that would impact the hydrolysis of β -lactams?*

Response: We thank the Reviewer for raising these important issues. The peptides from the PSM α family are not secreted in equal amounts during biofilm formation (e.g., Wang, R et al. Nat Med. 13, 1510–1514 (2007); Gordon Y.C et al., The *FASEB journal* 28 (2014)). It should be emphasized that the study of the roles and assembly properties of bacterially secreted functional amyloids is still in infancy, and as such not much is known about homopolymer and heteropolymer assemblies, and putative co-assembly of specific PSMs. Indeed, as recent reports have shown that multicomponent assembly of peptides may enhance catalytic efficiency of synthetic peptides (Das and coworkers “Short peptide-based cross- β amyloids exploit dual residues for phosphoesterase like activity”, *chemical Science* (2022)), further analysis of this very interesting aspect will be carried out in a future study.

2. Mention the overall charge of the peptide in Figure 2A.

Response: Done.

3. Does increasing the number of lysine residues in PSM α 2 or PSM α 3 increase their catalytic activity towards nitrocefin?

Response: This is indeed a key question. Although lysine residues are crucial for the catalytic activity and take a part in both binding of the β -lactam substrate and in the hydrolysis process itself, the primary factors affecting the reaction is the structure and position of the lysines rather than their number. This interpretation is supported by the following observations:

- Both PSM α 2 and PSM α 3 have four lysine residues, but their catalytic efficiency is significantly different.
- Mutants like α 3-(d)FF, α 3-F10P (Figure 3), α 2-(d)II and α 2-F10P (Figure S8) contain the same number of lysines (4), but differ in their structures compared to the parent peptide fibrils, resulting in much lower catalytic activities.
- The point mutations of PSMa3 (Figure 4) effectively eliminate a single lysine residue from the PSMa3 sequence, resulting in different activities.

4. *Polymorphism is common in protein aggregation. Is it known if the PSMs form polymorphic structures or do the aggregates formed by PMS peptides under different conditions have similar activity towards nitrocefin?*

Response: we thank the Reviewer for this comment. As the Reviewer points out, polymorphism is common in amyloids in general, and indeed polymorphism has been also reported in PSM α 3 in particular (see ref. 46). However, while PSM α 3 adopts both fibrillar and nanotubular structures, distinguishing between the catalytic activities of the species is experimentally challenging. This issue is further discussed and emphasized in the revised manuscript (pag 27, second paragraph).

5. *It is unclear if the authors tested a functional role for PSMs ability to hydrolyze β -lactams. Are *S. aureus* without PSMs more susceptible to β -lactam antibiotics? Have the authors tried any growth assay on *S. aureus* expressing each PSMa1-4 in the presence of antibiotics to validate the current finding.*

Response: These experiments will be informative. However, genetically engineered *S. aureus* strains not expressing PSMs, or expressing each PSM individually are yet to be produced. However, as indicated above (response to the first query), we carried out *S. aureus* growth experiments, demonstrating inhibition of antibiotic activity of beta lactams in the presence of PSMa3 amyloid fibrils.

REVIEWER COMMENTS

Reviewer #2 (Remarks to the Author):

In this revised manuscript by Arad et al, the authors aim to further convince of a kinetically meaningful and catalytically based clearance of beta-lactam antibiotics by *S. aureus* distinct PSM fibrils. They add two additional penicillin substrates beyond the initial nitrocefin, using LC-MS methods to show hydrolysis over several hours. They also add an MD analysis in combination with mutated lysine variants of the PSMa3 to predict possible localization of a cluster of 3 lysines to a putative modelled beta-lactam binding site. Unfortunately neither of these added analysis contribute to further significant mechanistic understanding of beta-lactam hydrolysis, nor clear observation that this very slow turn over of beta-lactam antibiotics is a major contributor of resistance in a physiological context.

The erratic nature of the curves (S2) and error in the nitrocefin (absorbance based) kinetic analysis (Figure 2D for eg) is high and so slow compared to beta-lactamases the significance and validity still remains unclear.

Other than suggesting lysines are important, little further mechanistic insight is possible from the MD simulation/docking and is also clouded further by the fact that the lysine mutants adopt different overall morphologies in the cryoTEM which could be an influence.

Given the very slow turnover, trapping a covalent intermediate with beta-lactam should be entirely possible, presumably then corroborating one of the lysines acting as a nucleophile.

There are still questions that would be critical for these to be true enzyme catalysts, including which residues would stabilize the generated oxyanion transition states. The other lysines? Suggesting a differential pKa amongst the lysine cluster? There are many electropositive polymers in nature, what are the features of these that promote catalysis?

In Figure 7, why does the addition of the PSMa3 fibrils (no nitrocefin; green curve) enhance *S. aureus* growth compared to *S. aureus* alone? Why are the errors in the PSMa3 with nitrocefin curve (blue) so much larger than the other curves?

Reviewer #3 (Remarks to the Author):

The authors have thoughtfully addressed my original comments and critiques. I just have a minor question and correction to add:

It would be good to add the growth curve of *S. aureus*. with PMSa1 or PMSa4 with and without nitrocefin in Figure 7 as a control. Why is an increase in the growth rate of *S. aureus*. in the presence of PMSa3 fibrils and PMSa3 fibrils + nitrocefin?

In Figure 4A there is something behind "sequence". It would be good to correct it.

Reviewer #4 (Remarks to the Author):

This is the revised version of the manuscript in which the authors described the helical phenol-soluble modulins (PSMs) of *Staphylococcus aureus*, called functional amyloids in this work, can degrade β -lactam antibiotics. I did not review the original manuscript, however, I agree that this is an important work, indicating that there is a newly discovered activity of PSMs. The kinetics of antibiotic degradation by PSMs is quite slow, however, the authors argue that these amyloid-like

molecules are less specific than classical beta lactamases, thus being potentially useful as antibiotic-resistance agents.

I have analyzed the comments and previous reviewers and answers of the authors. In my opinion, the authors addressed all critical comments. Some suggested experiments were not conducted, but they would be extremely time-consuming (for example, construction of strains and mutants which are not yet available, which is perhaps not an easy task, would be necessary), and the authors performed other, alternative experiments which provide answer to the asked questions. In summary, in my opinion the authors did a very good work, and the revised manuscript is acceptable for publication in Nature Communications.

Detailed response to Reviewers' comments

Reviewer #2:

*In this revised manuscript by Arad et al, the authors aim to further convince of a kinetically meaningful and catalytically based clearance of beta-lactam antibiotics by *S. aureus* distinct PSM fibrils. They add two additional penicillin substrates beyond the initial nitrocefin, using LC-MS methods to show hydrolysis over several hours. They also add an MD analysis in combination with mutated lysine variants of the PSMA3 to predict possible localization of a cluster of 3 lysines to a putative modelled beta-lactam binding site. Unfortunately neither of these added analysis contribute to further significant mechanistic understanding of beta-lactam hydrolysis, nor clear observation that this very slow turn over of beta-lactam antibiotics is a major contributor of resistance in a physiological context.*

Response:

We are grateful to the Reviewer for the evaluation of the revised manuscript. We appreciate the Reviewer's comments, however we do believe that the additional extensive experimental and computational analyses we included in the revised manuscript, in large part following the constructive comments of the Reviewer on the originally submitted manuscript, have contributed to considerably more comprehensive structural and mechanistic insight on the novel phenomenon we report - amyloid-catalyzed beta-lactam hydrolysis - further underlying the significance of our observations.

Concerning the very low turnover of the hydrolyzed beta lactams. As we specifically discuss in the manuscript, it is expected that the catalytic activity of the more promiscuous PSM α 3 amyloid fibrils will be significantly lower than β -lactamase enzymes, primarily accounting for the lesser degree of 'lock-key' specificity between the substrates and amyloid fibrils' surface, and less defined binding pocket on the fibril surface in comparison with classical enzymes. Indeed, to the best of our knowledge virtually all peptide aggregates studied thus far do exhibit significantly lower catalytic activities compared to biological enzymes. In that regard, the catalytic efficiency of PSM α 3 fibrils is relatively high in comparison to other catalytic peptides in general and amyloids in particular, which catalyze ester bond hydrolysis.

Concerning the possible contribution of PSMA3 amyloid catalysis of beta lactam hydrolysis in physiological contexts - it is conceivable that over longer timespans of antibiotics exposure to bacterial biofilms and their functional amyloid framework (which is the case in the real world in which bacteria are treated with antibiotics for days), catalytic degradation of antibiotic compounds may become significant. Indeed, please note the pronounced catalytic degradation of nitrocefin recorded in the presence of actual *S. aureus* biofilms (Figure 7B), an observation supporting this scenario.

Following the Reviewer's comment and to better account for the aspects discussed above, we further expanded the discussion of the activity difference between amyloids and enzymes (p. 29, top paragraph, in the revised manuscript).

The erratic nature of the curves (S2) and error in the nitrocefin (absorbance based) kinetic analysis (Figure 2D for eg) is high and so slow compared to beta-lactamases the significance and validity still remains unclear.

Response: The important feature in Figure S2 is the pronounced difference between the catalytic activity of the *S. aureus* – secreted functional amyloid on the one hand, and beta-amyloid, representing a non-bacterial human amyloid, on the other hand. To address the Reviewer’s comment regarding the shapes of the curves we carried out another experiment yielding smoother absorbance curves (revised Figure S2).

Concerning the large errors in Figure 2D – please note that these are not experimental errors, but rather the calculated confidence intervals obtained via the catalytic efficiency equation and thereby expected to be higher than measurement errors. Regardless, the key aspect in Figure 2 (and the bar diagram in panel 2D specifically) is the significant difference between the catalytic activities of PSM α 1 and PSM α 4, and PSM α 2/3.

Concerning the slow kinetics of the reaction compared to beta lactamases – please see our response to the first comment, above.

Other than suggesting lysines are important, little further mechanistic insight is possible from the MD simulation/docking and is also clouded further by the fact that the lysine mutants adopt different overall morphologies in the cryoTEM which could be an influence.

Response: We thank the Reviewer for highlighting the roles of lysines in the catalytic process. In fact, the analysis of Lys-Ala mutants (succinctly suggested by the Reviewer in the first review) and MD simulations underscore the key roles of the primary amine in the N-terminus and lysines 6, 9, 12 in binding of the β -lactam. Moreover, the important role of Lys-17 in the overall activity is also shown, yet not at the binding stage. Concerning the different overall morphologies of some of the mutants – this is correct. However, the important results in the context of the mutant catalytic activities are the fact that all mutants did assemble into amyloid aggregates, and adopted α -helical organization (i.e., FTIR and CD data in Figure S9) echoing the cross- α structure of the parent PSM α 3 fibrils.

Given the very slow turnover, trapping a covalent intermediate with beta-lactam should be entirely possible, presumably then corroborating one of the lysines acting as a nucleophile.

Response: We thank the Reviewer for this interesting mechanistic insight and proposition. Indeed, this structural / mechanistic aspect will be examined in a future study, as this work (which is already quite extensive) essentially presents the key experimental and computation features of the novel beta-lactam hydrolysis by functional bacterial amyloid phenomenon. Following the Reviewer’s comment, we added in the revised manuscript pertinent text referring to this structural possibility (p. 19, bottom).

There are still questions that would be critical for these to be true enzyme catalysts, including which residues would stabilize the generated oxyanion transition states. The other lysines? Suggesting a differential pKa amongst the lysine cluster? There are many electropositive polymers in nature, what are the features of these that promote catalysis?

Response: We thank the Reviewer for pointing out these issues. We would like to emphasize that the broad sequence library we analyzed, including four PSM α variants, ten PSM α 3 mutants, and three PSM α 2 mutants reveal several critical features of the amyloid-mediated beta-lactam catalysis: fibril formation, the cross- α structures, and presence of lysine arrays on the fibrils' surface. Indeed, differentiation on the pKa of the lysine residues may contribute to activation of the residues; Figure S10,A does show, however, no significant differentiation or shift in the measured Ka of the lysine residues in PSM α 3, suggesting that difference in the localized pKa is secondary to other effects, particularly the geometry of nitrocefin binding site upon the fibrils' surface (and the N-terminal region specifically).

Concerning transition state analysis which may shed light on oxyanion stabilization – this may indeed be informative however such a study, which should involve comprehensive quantum-mechanical calculations, is beyond the scope of this work and will be pursued in a follow-up study.

Indeed, there are many electropositive polymers in nature, but our study demonstrates that beta-lactam hydrolysis depends specifically on the molecular and structural properties of distinct members of the PSM α functional bacterial amyloid family and is not a general phenomenon encountered for any biological polymer.

*In Figure 7, why does the addition of the PSM α 3 fibrils (no nitrocefin; green curve) enhance *S. aureus* growth compared to *S. aureus* alone? Why are the errors in the PSM α 3 with nitrocefin curve (blue) so much larger than the other curves?*

Response: we thank the Reviewer for pointing out this experimental feature. The enhanced absorbance at $\lambda=600$ nm upon addition of PSM α 3 amyloid fibrils is probably attributed for higher turbidity induced by binding of the PSM α assemblies (which are cationic) to the bacterial cells that are rich in anionic polysaccharides (and concomitant aggregation). Similarly, the samples comprising PSM α 3 amyloid fibrils, bacteria, nitrocefin, and medium likely manifest greater variations of co-aggregates, contributing to the larger error bars observed. Following the Reviewer's comment and to better account for this experimental issue we added pertinent text (p. 24, last paragraph).

Reviewer #3:

The authors have thoughtfully addressed my original comments and critiques. I just have a minor question and correction to add:

We thank the Reviewer for this comment.

It would be good to add the growth curve of S. aureus. with PMSa1 or PMSa4 with and without nitrocefin in Figure 7 as a control.

Response: We thank the Reviewer for the constructive comment. Following this suggestion, we carried out these experiments and, indeed, PSM α 1 and PSM α 4 fibrils did not inhibit the toxic effect of nitrocefin (new Figure S15).

Why is an increase in the growth rate of S. aureus. in the presence of PMSa3 fibrils and PMSa3 fibrils + nitrocefin?

Response: Please see our response to the last comment by Reviewer #2, above.

In Figure 4A there is something behind “sequence”. It would be good to correct it.

Response: Done.

Reviewer #4:

This is the revised version of the manuscript in which the authors described the helical phenol-soluble modulins (PSM α s) of Staphylococcus aureus, called functional amyloids in this work, can degrade β -lactam antibiotics. I did not review the original manuscript, however, I agree that this is an important work, indicating that there is a newly discovered activity of PSM α s. The kinetics of antibiotic degradation by PSM α s is quite slow, however, the authors argue that these amyloid-like molecules are less specific than classical beta lactamases, thus being potentially useful as antibiotic-resistance agents.

I have analyzed the comments and previous reviewers and answers of the authors. In my opinion, the authors addressed all critical comments. Some suggested experiments were not conducted, but they would be extremely time-consuming (for example, construction of strains and mutants which are not yet available, which is perhaps not an easy task, would be necessary), and the authors performed other, alternative experiments which provide answer to the asked questions.

In summary, in my opinion the authors did a very good work, and the revised manuscript is acceptable for publication in Nature Communications.

We thank the Reviewer for thorough reading of the manuscript and our comments in response to the Reviewers' comments, and overall positive assessment of this work.

REVIEWERS' COMMENTS

Reviewer #3 (Remarks to the Author):

The authors addressed my comments. The comments from reviewer 2 might still need attention as the relatively low catalytic activity of the PSMs do raise unanswered questions as to the biological significance of the findings. Nonetheless, the beta-lactam degradation activity described in the manuscript is novel and interesting.